# QUEST: Quadruple Multimodal Contrastive Learning with Constraints and Self-Penalization

**Qi Song**[1]*, **Tianxiang Gong**[2]*, **Shiqi Gao**[2],
**Haoyi Zhou**[1,3]†, **Jianxin Li**[2,3]
[1]School of Software, Beihang University
[2]School of Computer Science and Engineering, Beihang University
[3]Zhongguancun Laboratory, Beijing
{songqi23, gongtx, gaoshiqi, haoyi, lijx}@buaa.edu

## Abstract

Multimodal contrastive learning (MCL) has recently demonstrated significant success across various tasks. However, the existing MCL treats all negative samples equally and ignores the potential semantic association with positive samples, which limits the model's ability to achieve fine-grained alignment. In multi-view scenarios, MCL tends to prioritize shared information while neglecting modality-specific unique information across different views, leading to feature suppression and suboptimal performance in downstream tasks. To address these limitations, we propose a novel contrastive framework named *QUEST: Quadruple Multimodal Contrastive Learning with Constraints and Self-Penalization*. In the QUEST framework, we propose quaternion contrastive objectives and orthogonal constraints to extract sufficient unique information. Meanwhile, a shared information-guided penalization is introduced to ensure that shared information does not excessively influence the optimization of unique information. Our method leverages quaternion vector spaces to simultaneously optimize shared and unique information. Experiments on multiple datasets show that our method achieves superior performance in multimodal contrastive learning benchmarks. On public benchmark, our approach achieves state-of-the-art performance, and on synthetic shortcut datasets, we outperform existing baseline methods by an average of $97.95\%$ on the CLIP model.

## 1 Introduction

Multimodal Contrastive Learning (MCL) has demonstrated robust representation capabilities and generalizability and effectively transferring to various downstream tasks (e.g. cross-modal retrieval [40, 39, 33], image captioning [37, 72, 73]). However, simply applying contrastive learning in multimodal scenarios presents significant challenges. ❶ In particular, contrastive learning treats all negative samples equally, ignoring the potential semantic relationships between negative samples and the anchor. ❷ Besides, current contrastive learning methods focus on maximizing mutual information between two views [68, 70] while ignoring unique information [41]. In multi-view scenarios, the assumption that modalities share substantial task-related information often does not hold, especially in complex datasets with minimal inter-modal overlap. ❸ Meanwhile, recent studies [56, 75] indicate that contrastive learning often neglects significant portions of input information, leading to feature suppression [1, 8] and shortcut learning [25, 57], where models minimize loss through the simplest path (e.g. shared information [3]), sacrificing deeper learning. These issues are prevalent in multimodal [45, 54, 32] and multi-view tasks [76, 82, 50, 43]. Recent approaches focus on pre-

---

*Equal contribution

†Corresponding author.

38th Conference on Neural Information Processing Systems (NeurIPS 2024).

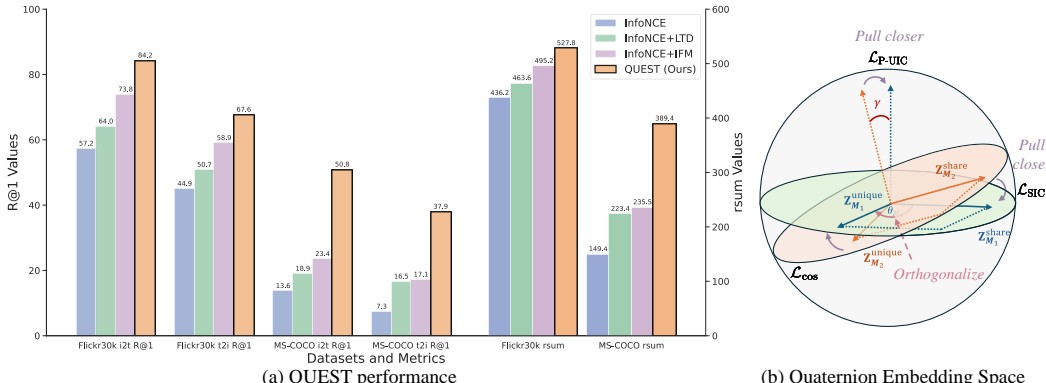

(a) QUEST performance          (b) Quaternion Embedding Space

Figure 1: (a) Our QUEST outperforms baselines $97.95\%$ on average when trained with task-related unique information and evaluated on downstream tasks on the CLIP model. (b) We build the quaternion embedding space, which aligns shared and unique representations from different modalities through the application of constraints and self-penalization. The $\mathcal{L}_{\text{SIC}}$ narrows the gap between shared representations, while $\mathcal{L}_{\text{P-UIC}}$ pulls the plane spanned by intra-modality shared and unique representations closer. Furthermore, Orthogonalization loss $\mathcal{L}_{cos}$ is employed to constrain the area.

serving more unique information, including reconstruction regularization [77, 4], implicit feature modification [58], and factorized representation [41], among others. However, these methods either overly introduce noise which may harm downstream tasks, or rely on certain assumptions (e.g., augmentation [41, 65, 45]). Additionally, we find that these methods do not explicitly distinguish unique information and still optimize using contrastive learning, making it difficult to avoid the model learning shortcuts [68, 29]. This raises the question: **can we explicitly extract both task-related unique and shared information without introducing too much noise?**

To this end, we proposes a novel contrastive framework called *QUEST: Quadruple Multimodal Contrastive Learning with Constraints and Self-Penalization*, designed to enhance the extraction and integration of both shared and unique information across multimodal data. Our primary motivation is to develop a mechanism that effectively captures unique information through a novel quaternion multimodal embedding space, as illustrated in Figure 1(b). This embedding space aims to pull shared representations closer while aligning the unique representations with the shared representation on a common plane. We achieve this by leveraging the properties of the normal vector from the cross-product to diversified unique representation. Consequently, our approach aligns commonalities across modalities while preserving the distinctive unique features of each modality.

Specifically, we first split a network into three components: an encoder, a shared decoder, and a unique decoder. The encoder learns general features with little bias toward specific tasks, while the shared decoder and unique decoder learn agreement and discriminative information, respectively. We build contrastive loss to constrain learning of shared information. To avoid the unique decoder degenerating into the shared decoder, we propose novel contrastive objectives and orthogonal constraints to optimize the quaternion vector space. Finally, self-penalization is used to prevent shared information from overly affecting quaternion vector space optimization. Our framework seeks to mitigate shortcut learning, offering a more nuanced, task-related learning paradigm. Our main contributions can be summarized as follows:

- We develop a novel framework to efficiently extract shared and unique information across multimodal data. To avoid the degeneration of the unique decoder, we propose an algorithm that utilizes quadruple embedding to constrain unique information from different views in a plane space.
- We consider that traditional CL overly relies on shared information due to data bias, causing failures with negative samples containing shared information related to the positive sample. Meanwhile, to prevent shared information from dominating the extraction of unique information, we introduce a self-penalization mechanism to dynamically reweight the distribution of negative samples, which penalizes hard negative samples. We provide theoretical analysis to show how this penalization effectively improves the extraction of unique information.
- We achieve state-of-the-art on popular datasets (e.g. MS-COCO [9] and Flickr30k [80]) compared to the baseline, demonstrating the general effectiveness of QUEST. Additionally, experiment results on synthetic shortcut datasets outperform baselines $97.95\%$ on average for CLIP, verifying the efficacy of QUEST.

# 2 Methods

## 2.1 Problem Formulation

For different modalities $\{\mathcal{M}_i\}_{i=1}^K$, given one modality, denoted as $\mathcal{M}_i$, along with its corresponding set of views $\{x_i^j\}_{j=1}^{N_i}$, $N_i \geq 1$, for one modality $\mathcal{M}_1$ encoder parameterized by $\Theta_1$, represented as $\mathcal{F}_{\mathcal{M}_1}(\cdot; \Theta_1)$, and another modality $\mathcal{M}_2$ encoder parameterized by $\Theta_2$, denoted as $\mathcal{F}_{\mathcal{M}_2}(\cdot; \Theta_2)$. These encoders process the sample from modal $\mathcal{M}_1$ and $\mathcal{M}_2$, for those modalities with multiple views, like modal one $\mathcal{M}_1$ and each of its views through their respective encoder, resulting in corresponding general representations $\mathbf{H}_{\mathcal{M}_1}^j = \mathcal{F}_{\mathcal{M}_1}(x_1^j; \Theta_1)$ and $\mathbf{H}_{\mathcal{M}_2}^j = \mathcal{F}_{\mathcal{M}_2}(x_2^j; \Theta_2)$. However, as illustrated in Figure 2, the InfoNCE loss maximizes task-related features shared across all modalities during training (i.e. $I(X_A'; X_B'; Y)$), while simultaneously suppressing the unique task-related features of each individual modality(i.e. $I(X_A; Y|X_B)$ and $I(X_B; Y|X_A)$). This process ultimately results in the loss of unique information. Therefore, the general representations are then separated into task-related shared and unique features through different decoders, i.e., the representations of different modalities $\mathbf{H}_{\mathcal{M}_1}^j, \mathbf{H}_{\mathcal{M}_2}^j$ are inputted separately into different decoders $\mathcal{G}_{\mathcal{M}_i}(\cdot)$ parameterized by $\Phi_i$. For the complete notations, refer to Appendix C, Table 6.

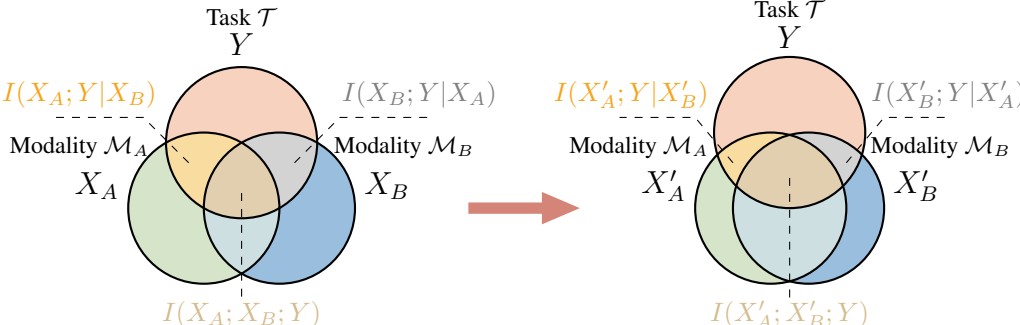

Figure 2: Feature suppression in multi-view contrastive learning. We define $I(X_A; X_B; Y)$ as task-related shared information, $I(X_A; Y|X_B)$ and $I(X_B; Y|X_A)$ as task-related unique information related to task $Y$ in modalities $X_A$ and $X_B$,respectively. Contrastive losses, such as InfoNCE, tend to maximize the task-related shared information while suppressing the task-related unique information in each modality. Left: before training with InfoNCE. Right: after training with InfoNCE.

## 2.2 Overview

In multi-view scenarios, the relationships between different modalities are many-to-many (e.g., for image retrieval, multiple captions can refer to the same image). Within a single modality, different views contain task-related unique and shared information. Additionally, task-related shared information may also exist among negative samples (as indicated by the red shading in Figure 3). Therefore, optimizing solely for shared information while ignoring unique information is suboptimal.

To address the challenge of overlooking unique information inherent to different perspectives in multimodal scenarios, we introduce the effective framework called *QUEST: Quadruple Multimodal Contrastive Learning with Constraints and Self-Penalization*. This framework extends existing contrastive learning methods by incorporating a four-partite architecture specifically designed to enhance the capture and integration of distinctive modal-specific features. The overall architecture is shown in Figure 3. According to [84, 62, 34], from the

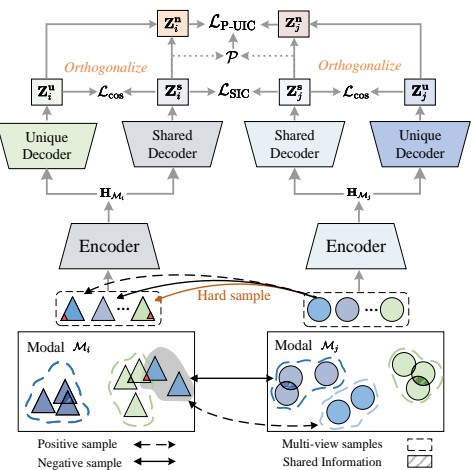

Figure 3: Framework of QUEST. The unique decoder is utilized to extra view-specific unique information and this process is guided by the proposed constraints and penalization.

perspective of neural network architecture, shallow layers learn low-level and general features while deeper layers learn task-biased high-level semantic features. Therefore, we define the encoder output as $H$, which contains shared $S$ and unique $U$ information related to the task, $H \supseteq S \cup U$. Consequently, we shared shallow layers for general representation to reduce computational cost and optimize the shared decoder and unique decoder for $S$ and $U$, respectively.

## 2.3 Quadruple InfoNCE

For each modality, the input data $\mathbf{X}_i$ (sampled from views) undergoes transformation by a modality-specific encoder $\mathcal{F}_{\mathcal{M}_i}(\cdot)$, producing an intermediary general representation denoted as $\mathbf{H}_{\mathcal{M}_i}$. Consequently, we introduce two decoders: a shared information decoder $\mathcal{G}^{\mathbf{s}}_{\mathcal{M}_i}(\cdot)$ and a unique information decoder $\mathcal{G}^{\mathbf{u}}_{\mathcal{M}_i}(\cdot)$. These decoders are tasked with disentangling the shared and unique components from the representation $\mathbf{H}_{\mathcal{M}_i}$, respectively. Let $\Theta_i$ represent the parameters of a shared encoder, while $\Phi_i$ and $\Psi_i$ symbolize the decoders for shared and unique information, respectively. The representation can be formulated as follows:

$$
\begin{aligned}
\mathbf{Z}^{\mathbf{u}}_i &= \mathcal{G}^{\mathbf{u}}_{\mathcal{M}_i}(\mathbf{H}_{\mathcal{M}_i}; \Phi_i) = \mathcal{G}^{\mathbf{u}}_{\mathcal{M}_i}(\mathcal{F}_{\mathcal{M}_i}(\mathbf{X}_i; \Theta_i); \Phi_i), \\
\mathbf{Z}^{\mathbf{s}}_i &= \mathcal{G}^{\mathbf{s}}_{\mathcal{M}_i}(\mathbf{H}_{\mathcal{M}_i}; \Psi_i) = \mathcal{G}^{\mathbf{s}}_{\mathcal{M}_i}(\mathcal{F}_{\mathcal{M}_i}(\mathbf{X}_i; \Theta_i); \Psi_i).
\end{aligned}
\tag{1}
$$

**Shared Information Constraint.** In multimodal and multi-view scenarios, data from different modalities and views often encapsulate shared information vital for model training. We conduct a shared Information Constraint (SIC) to maximize the lower bound of MI between representations from different views to encourage the shared decoder to learn agreement related to the task. This constraint is optimized by computing the InfoNCE loss between ($\mathbf{Z}^{\mathbf{s}}_i$ and $\mathbf{Z}^{\mathbf{s}}_j$), defined as,

$$
\mathcal{L}_{\text{SIC}} = \sum_{i,j} \mathbb{1}_{\mathcal{M}_i \neq \mathcal{M}_j} \mathbb{E}_{\mathbf{z}^{\mathbf{s}}_i} \left[ -\log \frac{\exp(s(\mathbf{Z}^{\mathbf{s}}_i, \mathbf{Z}^{\mathbf{s}+}_j)/\tau)}{\exp(s(\mathbf{Z}^{\mathbf{s}}_i, \mathbf{Z}^{\mathbf{s}+}_j)/\tau) + \sum_{k=1}^{m} \mathbb{1}_{\hat{\mathbf{y}}^-} \exp(s(\mathbf{Z}^{\mathbf{s}}_i, \mathbf{Z}^{\mathbf{s}-}_{jk})/\tau)} \right]. \tag{2}
$$

**Unique Information Constraint.** In contrast to shared information, unique information is modality-specific and task-related, providing essential insights for downstream tasks. To preserve this, we introduce a Unique Information Constraint (UIC), extracting the unique information that exists within different views, which are unrelated to each other yet relevant to the task. Relying solely on Shared Information Constraint (SIC) is insufficient to preserve this information [41, 45, 3]. Strict constraints, such as directly enforcing consistency of distributions across diverse views, may lead to the Unique Information Constraint converging to the SIC, particularly in scenarios with identical input representations, loss functions, and network structures. To address this issue, we implement a less stringent constraint, aiming to maximize the similarity between the spaces formed by unique and shared information across different modalities. Firstly, we derive the representation space of normal vectors for shared and unique embedding spaces through cross-product calculations,

$$
\mathbf{Z}^{\mathbf{n}}_i = \mathbf{Z}^{\mathbf{s}}_i \times \mathbf{Z}^{\mathbf{u}}_i. \tag{3}
$$

In the newly projected space, our objectives aim to maximize the alignment of unique representation from different modalities within the plane spanned by the shared representation, the magnitude can be formulated as:

$$
(\mathbf{Z}^{\mathbf{s}}_i \times \mathbf{Z}^{\mathbf{u}}_i) \cdot (\mathbf{Z}^{\mathbf{s}}_j \times \mathbf{Z}^{\mathbf{u}}_j) = \|\mathbf{Z}^{\mathbf{s}}_i\| \|\mathbf{Z}^{\mathbf{u}}_i\| \sin\alpha \|\mathbf{Z}^{\mathbf{s}}_j\| \|\mathbf{Z}^{\mathbf{u}}_j\| \sin\beta \cos\gamma, \tag{4}
$$

where $\sin\alpha$ and $\sin\beta$ represent the sine similarity between the shared and unique representation across different modalities, and $\cos\gamma$ represents the cosine similarity of the normal vector. We maximize $\sin\alpha$ and $\sin\beta$ via orthogonalized cosine loss $\mathcal{L}_{\cos}$ and contrastive loss to maximize $\cos\gamma$, the unique information constraint can be formulated as:

$$
\begin{aligned}
\mathcal{L}_{\text{UIC}} = &\sum_{i,j} \mathbb{1}_{\mathcal{M}_i \neq \mathcal{M}_j} \mathbb{E}_{\mathbf{z}^{\mathbf{n}}_i} \left[ -\log \frac{\exp(s(\mathbf{Z}^{\mathbf{n}}_i, \mathbf{Z}^{\mathbf{n}^+}_j)/\tau)}{\exp(s(\mathbf{Z}^{\mathbf{n}}_i, \mathbf{Z}^{\mathbf{n}^+}_j)/\tau) + \sum_{k=1}^{m} \mathbb{1}_{\hat{\mathbf{y}}^-} \exp(s(\mathbf{Z}^{\mathbf{n}}_i, \mathbf{Z}^{\mathbf{n}^-}_{jk})/\tau)} \right] \\
&+ \underbrace{\sum_i \sum_j \frac{\mathbf{Z}^{\mathbf{s}}_{ij} \cdot \mathbf{Z}^{\mathbf{u}}_{ij}}{\|\mathbf{Z}^{\mathbf{s}}_{ij}\| \|\mathbf{Z}^{\mathbf{u}}_{ij}\|}}_{\mathcal{L}_{\cos}}.
\end{aligned}
\tag{5}
$$

By pulling positive samples closer and pushing negative samples away, we constrain the shared and unique representations of different modalities to coexist within the same spatial plane as much as practicable. Concurrently, we employ $\mathcal{L}_{\text{cos}}$ to maintain the diversity of unique information and ensure that $\mathbf{Z}_i^{\mathbf{s}} \neq \mathbf{Z}_i^{\mathbf{u}}$. It is worth noting that we do not impose any constraints on the unique representations of different modalities, as these may be inherently unrelated.

**Comparison.** Applying standard InfoNCE to extract unique information leads to suboptimal generalization, as demonstrated in Section 3.3. To address this, we propose UIC with indirect vectors $\mathbf{Z}^n$, where the cross-product operation fundamentally alters gradient propagation patterns compared to pair-wised InfoNCE. Specifically, given $\mathcal{L}_{\text{InfoNCE}} = -\log \frac{\exp(Z_a \cdot Z_b^+/\tau)}{\sum_{i=0}^N \exp(Z_a \cdot Z_i/\tau)}$ [70, 83], the gradient with respect to anchor embedding $Z_a$ as follows,

$$-\frac{\partial \mathcal{L}_{\text{InfoNCE}}}{\partial Z_a} = \frac{1}{\tau}(Z_b^+ - \sum_{i=0}^N \beta_i Z_{bi}), \qquad (6)$$

where $\beta_i = \frac{\exp(Z_a \cdot Z_i/\tau)}{\sum_{i=0}^N \exp(Z_a \cdot Z_i/\tau)}$ and $Z_b^+$ is random sampled from positive set $\{Z_b^1, ...Z_b^j\}$. Under the assumption that different views hold both shared and unique information, we have $I(Z_b^j; Y) = I(Z_b^j; Z_a; Y) + I(Z_b^j; Y | Z_a)$, where $I(Z_b^j; Z_a; Y)$ represents shared information between two views and $I(Z_b^j; Y | Z_a)$ represents unique information for $j$th view. With sufficient training iterations, the unique information tends towards noise as shared information dominates the accumulated gradient (first term in Eq. (6)). This is consistent with the conclusion of MI [41]. Assume that the features obtained from the encoder consist of shared features $S$ which correspond to the anchor and unique features $U$, represented as $Z_b^j = (S \cup U^j)$. Traditional contrastive learning defines an additive model $Z_b^j = (S + U^j)$ whereas $Z_b^j = (S \times U^j)$ in our model. Intuitively, there exists $\zeta$ satisfes $\zeta = Z_a \cdot (S \times U^1) = ... = Z_a \cdot (S \times U^j)$. Therefore, UIC is a weaker constraint that ensures quaternion vectors between different views lie on the same plane as much as possible. If we use both SIC and UIC simultaneously, SIC will pull the shared representations of different views closer, while UIC will ensure that the unique representations of different views lie on the same plane as much as possible, rather than measuring their cosine similarity, as unique information is uncorrelated. We conducted extensive experiments to verify this (Section 3.3 for more details).

## 2.4 Shared Information Guided Constraint

Contrastive learning fundamentally operates by optimizing vector representations to minimize distances between positive pairs while maximizing distances between negative pairs in the embedding space. However, a critical limitation inherent in conventional approaches stems from their undifferentiated treatment of all samples within a batch $\mathcal{B}$ as negative examples. This indiscriminate categorization may inadvertently cause the model to overlook potential semantic relationships (as illustrated by the red shading in Figure 3), despite their shared semantic content (e.g., different image captions containing identical substrings in image-text retrieval tasks). Such misclassification significantly impairs the model's representation capacity. We refer to these cases as "hard negative samples. While existing methods attempt to address this issue through clustering-based approaches, their two-stage nature limits practical adoption. Leveraging our dual-branch architecture, we propose a more effective solution that directly utilizes the output of the shared decoder as a supervision signal for the penalty term.

Considering our objectives is to optimize the shared information decoder through $\mathcal{L}_{\text{SIC}}$ and unique information decoder through $\mathcal{L}_{\text{UIC}}$, and the shared information also affects the optimization of unique information as shown in Eq. (4), we attempt to use the intra-model shared information similar to penalization to guide the optimization of unique information. Unlike soft label [49, 17, 16, 47, 60, 24] which aim to mitigate the strict constraints of one-hot labels, preventing overconfidence by retaining more potential positive samples, our method aims to impose stricter constraints to suppress shared information in the process of learning unique information. Specifically, the more shared information between the anchor and all the negative samples, the greater the encouragement in learning unique information between the anchor and the positive sample. Formally, for shared representation $\mathbf{Z}_i^{\text{s}}$ and $\mathbf{Z}_j^{\text{s}}$, the weighted similarity matrix can be formulated as:

$$\mathcal{P} = \exp(\lambda[\mathbf{S} - \text{diag}(\mathbf{S}) + \mathbf{I}]), \qquad (7)$$

where $\mathbf{S} = \mathbf{Z}_i^{\mathbf{s}}\mathbf{Z}_j^{\mathbf{s}\,T}$ is the similarity matrix, $\mathbf{I}$ is identity matrix, $\lambda \in \mathbb{R}$ is learnable weight parameter. Next, the weighted similarity matrix $\mathcal{P} \in \mathbb{R}^{N \times N}$ is utilized as penalization to supervise the optimization of unique information satisfied $\mathcal{P}_{ij,i \neq j} \propto s(z_i^s, z_j^s)$, the penalized UIC can be defined as:

$$
\mathcal{L}_{\text{P-UIC}} = \sum_{i,j} \mathbb{1}_{\mathcal{M}_i \neq \mathcal{M}_j} \mathbb{E}_{\mathbf{Z}_i^{\mathbf{n}}} \left[ -\log \frac{\exp(s(\mathbf{Z}_i^{\mathbf{n}}, \mathbf{Z}_j^{\mathbf{n}^+})/\tau)}{\exp(s(\mathbf{Z}_i^{\mathbf{n}}, \mathbf{Z}_j^{\mathbf{n}^+})/\tau) + \sum_{k=1}^{m} \mathbb{1}_{\hat{\mathbf{y}}^-} \exp(\mathcal{P}_k \cdot s(\mathbf{Z}_i^{\mathbf{n}}, \mathbf{Z}_{jk}^{\mathbf{n}^-})/\tau)} \right]
$$
$$
+ \sum_i \mathcal{L}_{\text{cos}}. \tag{8}
$$

**Gradient Analysis.** For simplicity, we set $\lambda = \sum_{k=0}^{m} \exp(\mathcal{P}_k \cdot s(\mathbf{Z}_i^{\mathbf{n}}, \mathbf{Z}_{jk}^{\mathbf{n}})/\tau)$ and ignore the second term, we reformulate Eq. (8) as:

$$
\widetilde{\mathcal{L}}_{\text{P-UIC}} = \mathbb{E}_{\mathbf{Z}_i^{\mathbf{n}}} \left[ -\log \frac{\exp(\mathcal{P}^+ \cdot s(\mathbf{Z}_i^{\mathbf{n}}, \mathbf{Z}_j^{\mathbf{n}^+})/\tau)}{\exp(\mathcal{P}^+ \cdot s(\mathbf{Z}_i^{\mathbf{n}}, \mathbf{Z}_j^{\mathbf{n}^+})/\tau) + \sum_{k=1}^{m} \mathbb{1}_{\hat{\mathbf{y}}^-} \exp(\mathcal{P}_k \cdot s(\mathbf{Z}_i^{\mathbf{n}}, \mathbf{Z}_{jk}^{\mathbf{n}^-})/\tau)} \right]
$$
$$
= \mathbb{E}_{\mathbf{Z}_i^{n}} \left[ \log \lambda - \frac{\mathcal{P}^+ \cdot s(\mathbf{Z}_i^{\mathbf{n}}, \mathbf{Z}_j^{\mathbf{n}^+})}{\tau} \right]. \tag{9}
$$

The gradient can be calculated as (Appendix D.2 for details):

$$
-\frac{\partial \widetilde{\mathcal{L}}_{\text{P-UIC}}}{\partial \mathbf{Z}_i^{\mathbf{n}}} = \frac{\mathcal{P}^+}{\tau} \frac{\partial s^+}{\partial \mathbf{Z}_i^{\mathbf{n}}} - \frac{1}{\lambda \tau} \sum_{k=0}^{m} \mathcal{P}_k \exp\left(\frac{\mathcal{P}_k s^k}{\tau}\right) \frac{\partial s^k}{\partial \mathbf{Z}_i^{\mathbf{n}}}, \tag{10}
$$

where $s^+ = s(\mathbf{Z}_i^{\mathbf{n}}, \mathbf{Z}_j^{\mathbf{n}^+})$ and $s^k = s(\mathbf{Z}_i^{\mathbf{n}}, \mathbf{Z}_{jk}^{\mathbf{n}^-})$. Intuitively, $\mathcal{P}$ represents the belief mass based on shared information, and the larger $P_k$ indicates more shared information between hard positive samples, which also influences the optimization of the unique decoder as in Eq. (4). When $\mathcal{P}_k = 0$ (i.e, $s(\mathbf{Z}_i, \mathbf{Z}_{jk}) \approx 0$), it will degenerate to the original InfoNCE loss. For hard negatives samples hold shared information where $s(\mathbf{Z}_i^{\mathbf{s}}, \mathbf{Z}_j^{\mathbf{s}}) > 0$, we increase the gradient using a penalty term which ensures that even if $\mathbf{Z}_i^{\mathbf{s}}$ is relatively large in Eq. (4), the lower loss reinforce other terms to constrain the overall value. From mutual information perspectives (Appendix D.3 for details), we have

$$
I(Z_i, Z_j) \geq H^{\tilde{P}}(Z_j|Z_i) - H(Z_j|Z_i) + \log N - \widetilde{\mathcal{L}}_{\text{P-UIC}}. \tag{11}
$$

When all negative samples $k$ satisfy $s(\mathbf{Z}_i, \mathbf{Z}_{jk}) \approx 0$, we obtain $H^{\tilde{P}}(Z_j|Z_i) = H(Z_j|Z_i)$. Subsequently, as the shared information between hard positive samples increases, $H^{\tilde{P}}(Z_j|Z_i)$ correspondingly increases, thereby elevating both the lower bound of mutual information and the confidence level, which consequently facilitates the learning of unique information.

## 2.5 Training Objectives

In summary, we apply (1) SIC (Eq. 2) to keep shared information relevance between different modalities, (2) UIC (Eq. 5) to build quadruple embedding space by maximizing the alignment of normal vector spanned by shared representation and unique representation, and the shared information is jointly optimized by SIC and UIC, (3) Self-penalization (Eq. 7) to amplify the effect of false positive samples in unique information optimization. The overall objective can be formulated as:

$$
\mathcal{L}_{\text{QUEST}} = \mathcal{L}_{\text{SIC}} + \mathcal{L}_{\text{P-UIC}}. \tag{12}
$$

# 3 Experiment

## 3.1 Experiment Setup

**Baselines and Setup.** Shortcut learning refers to the process in deep learning model training where the model completes tasks (such as classification, retrieval, etc.) by learning simple and discriminatory features while ignoring the semantic and more complex features of the data. This can result in poor model performance on downstream tasks. Latent target decoding (LTD) [4],

and implicit feature modification (IFM) [58] are two methods that mitigate shortcut learning by reducing feature suppression. LTD reconstructs the caption representations in the latent space of a Sentence-BERT model, allowing the encoder to mitigate feature suppression via correct mapping. IFM perturbs discriminatory features through encoders and removes part of these features to avoid learning shortcuts, which is implemented as a dual loss combined with the InfoNCE loss. We provide source code of our paper. [2].

Image Caption Retrieval(ICR) retrieves the most relevant sample in another modality by using a sample of one modality as a query. In this task, there are two retrieval modes: text-to-image (t2i) and image-to-text (i2t). We evaluated our method in the ICR task using CLIP [52] and VSE++ [23] models on Flickr30k [80] and MS-COCO [42] datasets.

Bleeker et al. [3] proposed synthetic shortcuts for the vision-language framework. This allows us to evaluate whether vision language (VL) models capture easy-to-learn discriminatory features or task-related information. We add MNIST Images to the top of pictures and appending corresponding numbers at the end of their respective captions. This controlled approach preserves the original information from both modalities and increases explicit mutual information between them.

**Evaluation Metrics.** To evaluate the model's performance on the Flickr30k and MS-COCO-Caption datasets, the Recall@K (i.e. R@1, R@5, R@10, which refers to the proportion of instances where the correct answer appears among the top K returned results out of all instances) and recall sum (RSUM) were selected as evaluation metrics for both i2t and t2i retrieval.

**Implementation Details.** We select ViT-B/32 as the visual backbone for CLIP and resnet152 for the VSE++ model when we evaluate them on the caption retrieval task using Flickr30k and MS-COCO dataset. We fine-tuned the pre-trained CLIP on downstream tasks and trained VSE++ from scratch with our method.

**Alternatives of Unique Decoder.** The key of multi-view assumption lies in the extraction of unique information, current methods achieve this by employing a single-layer MLP, ensuring orthogonality with shared information or through data augmentation and factorized loss. In our framework, a single-layer MLP is used for the ResNet backbone, while a two-layer Transformer is implemented for the Transformer backbone. Detailed methodologies are provided in the Appendix B.2.

## 3.2 Performance Evaluation

**QUEST vs. Vanilla InfoNCE.** QUEST outperforms the vanilla InfoNCE, as shown in Table 1. On the Flickr30k test set, QUEST yields $R@1$ improvements of $(2.4, 1.5)$ for CLIP and $(1.1, 3.4)$ for VSE++ in i2t and t2i tasks.. The corresponding RSUM metrics increase by $3.2$ and $12.1$. On MS-COCO, QUEST achieves $R@1$ gains of $(1.6, 0.9)$ for CLIP and $(3.1, 3.2)$ for VSE++ in i2t and t2i tasks, with corresponding RSUM improvements of $8.1$ and $17.3$. Additionally, QUEST exhibits faster convergence to the optimal solution.

**Experiment on Synthetic Shortcuts.** To assess the effectiveness of our proposed QUEST mitigating feature suppression in Contrastive Learning, we use synthetic shortcuts [3] by injecting easy-to-learn and discriminatory shared information into the image-text training dataset. We then evaluate the model's performance on downstream tasks with and without these synthetic shortcuts to determine if the presence of shortcuts causes the suppression of other task-related information and an over-reliance on shortcut features. Our baselines' results are consistent with [3].

As shown in Table 1, adding shortcuts leads to performance degradation across all models to some degree, indicating that the models have not learned sufficient shared and unique information. However, our method outperforms LTD and IFM, indicating it captures task-related information more effectively in downstream tasks (evaluation without shortcuts).

**CLIP Performance Enhancement.** Our method significantly enhances the CLIP model's performance on the Flickr30k and MS-COCO datasets. Compared to InfoNCE, we observe substantial R@1 improvements in both i2t and t2i tasks, with RSUM increases of $91.6$ and $240$ on Flickr30k and MS-COCO, respectively. Our approach also outperforms previous SOTA methods, surpassing $\mathcal{L}_{\text{InfoNCE+IFM}}$ on Flickr30k and showing sinificant performance improvements against $\mathcal{L}_{\text{InfoNCE}}$ on MS-COCO.

---

[2] https://github.com/Vortexsong/QUEST

Table 1: Result on Flickr30k and MS-COCO with varied method. sc denotes shortcut, we evaluate CLIP and VSE++ w/wo shortcut on i2t and i2i task. QUEST outperforms InfoNCE and achieve superior performance compare with other baselines in most cases. †denote use of ltd.

| Method | sc | Flickr30k | | | | | | | MS-COCO | | | | | | |
|---|---|---|---|---|---|---|---|---|---|---|---|---|---|---|---|
| | | i2t | | | t2i | | | RSUM | i2t | | | t2i | | | RSUM |
| | | R@1 | R@5 | R@10 | R@1 | R@5 | R@10 | | R@1 | R@5 | R@10 | R@1 | R@5 | R@10 | |
| **CLIP** | | | | | | | | | | | | | | | |
| $\mathcal{L}_{\text{InfoNCE}}$ | ✗ | $86.9_{\pm0.1}$ | $97.4_{\pm0.1}$ | $99.0_{\pm0.0}$ | $72.4_{\pm0.1}$ | $92.1_{\pm0.0}$ | $95.8_{\pm0.0}$ | $543.5_{\pm1.1}$ | $63.8_{\pm0.3}$ | $86.1_{\pm0.0}$ | $92.3_{\pm0.0}$ | $46.3_{\pm0.3}$ | $74.8_{\pm0.1}$ | $84.1_{\pm0.2}$ | $447.5_{\pm0.5}$ |
| $\mathcal{L}_{\text{InfoNCE+LTD}}$ | ✗ | $86.5_{\pm0.6}$ | $97.1_{\pm0.0}$ | $98.5_{\pm0.0}$ | $72.4_{\pm0.0}$ | $\mathbf{92.3}_{\pm0.0}$ | $\mathbf{95.9}_{\pm0.0}$ | $542.8_{\pm0.8}$ | $63.8_{\pm0.0}$ | $86.1_{\pm0.0}$ | $92.3_{\pm0.0}$ | $46.3_{\pm0.0}$ | $74.7_{\pm0.0}$ | $84.1_{\pm0.0}$ | $447.4_{\pm0.0}$ |
| $\mathcal{L}_{\text{InfoNCE+IFM}}$ | ✗ | $87.4_{\pm0.1}$ | $97.4_{\pm0.2}$ | $99.1_{\pm0.0}$ | $73.2_{\pm0.0}$ | $92.2_{\pm0.0}$ | $95.6_{\pm0.0}$ | $544.9_{\pm0.2}$ | $63.0_{\pm0.1}$ | $86.6_{\pm0.1}$ | $92.6_{\pm0.2}$ | $47.2_{\pm0.0}$ | $75.6_{\pm0.0}$ | $84.5_{\pm0.0}$ | $449.5_{\pm1.7}$ |
| $\mathcal{L}_{\text{QUEST(Ours)}}$ | ✗ | $\mathbf{89.3}_{\pm0.3}$ | $\mathbf{97.8}_{\pm0.2}$ | $\mathbf{99.2}_{\pm0.3}$ | $\mathbf{73.9}_{\pm0.3}$ | $91.5_{\pm0.3}$ | $95.0_{\pm0.3}$ | $\mathbf{546.7}_{\pm1.9}$ | $\mathbf{65.4}_{\pm0.1}$ | $\mathbf{87.7}_{\pm0.2}$ | $\mathbf{93.6}_{\pm0.4}$ | $\mathbf{48.5}_{\pm0.2}$ | $\mathbf{75.7}_{\pm0.5}$ | $\mathbf{84.7}_{\pm0.6}$ | $\mathbf{455.6}_{\pm2.4}$ |
| $\mathcal{L}_{\text{InfoNCE}}$ | ✓ | $57.2_{\pm8.3}$ | $84.0_{\pm4.8}$ | $91.0_{\pm1.9}$ | $44.9_{\pm4.5}$ | $74.9_{\pm6.0}$ | $84.2_{\pm2.5}$ | $436.2_{\pm145.0}$ | $13.6_{\pm0.9}$ | $31.5_{\pm2.4}$ | $42.2_{\pm3.7}$ | $7.3_{\pm0.6}$ | $22.1_{\pm1.0}$ | $32.7_{\pm1.7}$ | $149.4_{\pm32.7}$ |
| $\mathcal{L}_{\text{InfoNCE+LTD}}$ | ✓ | $64.0_{\pm1.3}$ | $87.8_{\pm0.9}$ | $93.2_{\pm0.8}$ | $50.7_{\pm0.6}$ | $79.8_{\pm0.7}$ | $88.1_{\pm0.5}$ | $463.6_{\pm17.3}$ | $18.9_{\pm0.1}$ | $41.8_{\pm0.1}$ | $54.1_{\pm0.1}$ | $16.5_{\pm0.0}$ | $39.4_{\pm0.0}$ | $52.6_{\pm0.1}$ | $223.4_{\pm0.2}$ |
| $\mathcal{L}_{\text{InfoNCE+IFM}}$ | ✓ | $73.8_{\pm0.8}$ | $91.5_{\pm0.5}$ | $95.6_{\pm0.0}$ | $58.9_{\pm0.1}$ | $84.4_{\pm0.1}$ | $91.1_{\pm0.2}$ | $495.2_{\pm5.7}$ | $23.4_{\pm1.5}$ | $46.5_{\pm2.7}$ | $58.2_{\pm2.5}$ | $17.1_{\pm0.3}$ | $38.9_{\pm0.9}$ | $51.3_{\pm1.0}$ | $235.5_{\pm43.8}$ |
| $\mathcal{L}_{\text{QUEST(Ours)}}$ | ✓ | $\mathbf{84.2}_{\pm0.3}$ | $\mathbf{96.0}_{\pm0.1}$ | $\mathbf{97.7}_{\pm0.2}$ | $\mathbf{67.6}_{\pm0.5}$ | $\mathbf{88.9}_{\pm0.2}$ | $\mathbf{93.4}_{\pm0.1}$ | $\mathbf{527.8}_{\pm1.4}$ | $\mathbf{50.8}_{\pm0.3}$ | $\mathbf{75.4}_{\pm0.4}$ | $\mathbf{84.1}_{\pm0.4}$ | $\mathbf{37.9}_{\pm0.3}$ | $\mathbf{65.1}_{\pm0.3}$ | $\mathbf{76.1}_{\pm0.4}$ | $\mathbf{389.4}_{\pm2.1}$ |
| **VSE++** | | | | | | | | | | | | | | | |
| $\mathcal{L}_{\text{InfoNCE}}$ | ✗ | $52.6_{\pm1.1}$ | $79.8_{\pm0.1}$ | $87.8_{\pm0.1}$ | $39.5_{\pm0.3}$ | $69.8_{\pm0.0}$ | $79.4_{\pm0.1}$ | $409.0_{\pm4.0}$ | $42.2_{\pm0.1}$ | $72.7_{\pm0.1}$ | $83.2_{\pm0.1}$ | $30.9_{\pm0.0}$ | $61.2_{\pm0.1}$ | $73.5_{\pm0.1}$ | $363.8_{\pm2.3}$ |
| $\mathcal{L}_{\text{InfoNCE+LTD}}$ | ✗ | $54.1_{\pm0.1}$ | $81.1_{\pm0.8}$ | $88.6_{\pm0.1}$ | $42.5_{\pm0.0}$ | $71.9_{\pm0.1}$ | $81.3_{\pm0.0}$ | $419.6_{\pm0.1}$ | $43.6_{\pm0.1}$ | $73.5_{\pm0.0}$ | $83.7_{\pm0.0}$ | $32.4_{\pm0.1}$ | $62.5_{\pm0.0}$ | $74.7_{\pm0.0}$ | $370.5_{\pm0.1}$ |
| $\mathcal{L}_{\text{InfoNCE+IFM}}$ | ✗ | $52.4_{\pm0.2}$ | $76.9_{\pm0.1}$ | $85.3_{\pm0.0}$ | $39.1_{\pm0.0}$ | $68.8_{\pm0.1}$ | $78.2_{\pm0.1}$ | $400.7_{\pm0.0}$ | $40.2_{\pm0.0}$ | $70.8_{\pm0.1}$ | $81.6_{\pm0.1}$ | $30.8_{\pm0.0}$ | $61.5_{\pm0.0}$ | $74.3_{\pm0.0}$ | $359.3_{\pm1.1}$ |
| $\mathcal{L}_{\text{QUEST(Ours)}}$ | ✗ | $\mathbf{54.7}_{\pm0.2}$ | $\mathbf{81.3}_{\pm0.4}$ | $\mathbf{88.8}_{\pm0.3}$ | $\mathbf{42.9}_{\pm0.1}$ | $\mathbf{72.3}_{\pm0.4}$ | $\mathbf{81.6}_{\pm1.1}$ | $\mathbf{421.6}_{\pm2.5}$ | $\mathbf{45.3}_{\pm0.1}$ | $\mathbf{75.5}_{\pm0.5}$ | $\mathbf{85.4}_{\pm0.1}$ | $\mathbf{34.1}_{\pm0.1}$ | $\mathbf{64.5}_{\pm0.2}$ | $\mathbf{76.3}_{\pm0.2}$ | $\mathbf{381.1}_{\pm1.5}$ |
| $\mathcal{L}_{\text{InfoNCE}}$ | ✓ | $0.1_{\pm0.0}$ | $0.4_{\pm0.0}$ | $0.8_{\pm0.0}$ | $0.1_{\pm0.0}$ | $0.4_{\pm0.0}$ | $1.0_{\pm0.0}$ | $2.9_{\pm0.0}$ | $0.0_{\pm0.0}$ | $0.1_{\pm0.0}$ | $0.2_{\pm0.0}$ | $0.0_{\pm0.0}$ | $0.1_{\pm0.0}$ | $0.2_{\pm0.0}$ | $0.6_{\pm0.0}$ |
| $\mathcal{L}_{\text{InfoNCE+LTD}}$ | ✓ | $24.7_{\pm0.5}$ | $\mathbf{51.8}_{\pm0.7}$ | $\mathbf{65.6}_{\pm1.4}$ | $\mathbf{20.7}_{\pm1.0}$ | $\mathbf{49.2}_{\pm0.6}$ | $\mathbf{62.6}_{\pm1.2}$ | $\mathbf{274.6}_{\pm4.6}$ | $3.9_{\pm0.0}$ | $13.7_{\pm0.6}$ | $21.6_{\pm0.9}$ | $3.1_{\pm0.2}$ | $11.0_{\pm1.6}$ | $18.1_{\pm3.0}$ | $71.4_{\pm3.6}$ |
| $\mathcal{L}_{\text{InfoNCE+IFM}}$ | ✓ | $0.0_{\pm0.0}$ | $0.6_{\pm0.1}$ | $0.9_{\pm0.2}$ | $0.1_{\pm0.0}$ | $0.5_{\pm0.0}$ | $1.0_{\pm0.0}$ | $3.2_{\pm0.0}$ | $0.0_{\pm0.0}$ | $0.1_{\pm0.0}$ | $0.2_{\pm0.0}$ | $0.0_{\pm0.0}$ | $0.1_{\pm0.0}$ | $0.2_{\pm0.0}$ | $0.7_{\pm0.0}$ |
| $\mathcal{L}_{\text{QUEST(Ours)}}$† | ✓ | $\mathbf{24.9}_{\pm0.4}$ | $48.4_{\pm0.3}$ | $61.1_{\pm0.5}$ | $17.5_{\pm0.3}$ | $43.4_{\pm0.6}$ | $56.5_{\pm0.8}$ | $251.8_{\pm2.9}$ | $\mathbf{10.5}_{\pm0.6}$ | $\mathbf{27.9}_{\pm0.3}$ | $\mathbf{40.6}_{\pm0.9}$ | $\mathbf{9.4}_{\pm0.5}$ | $\mathbf{29.0}_{\pm1.4}$ | $\mathbf{42.6}_{\pm2.1}$ | $\mathbf{160.0}_{\pm5.8}$ |

**Comparative Analysis of VSE++ Improvements.** VSE++ also shows improvements on both datasets, with increases in $R@1$ for i2t and t2i tasks compared to $\mathcal{L}_{\text{InfoNCE}}$. On MS-COCO, VSE++ outperforms $\mathcal{L}_{\text{InfoNCE+IFM}}$, enhancing $R@1$ and $rsum$ metrics. However, $\mathcal{L}_{\text{InfoNCE+LTD}}$ achieves better performance on Flickr30k, excelling in text modality reconstruction and enhancement. This discrepancy may be attributed to VSE++'s GRU text encoder, which appears to be less effective in capturing modality-independent information on smaller datasets like Flickr30k .

## 3.3 Ablation Study

We also evaluate the effects of two constraints (shared information constraint $\mathcal{L}_{\text{SIC}}$ and unique information constraint $\mathcal{L}_{\text{UIC}}$) and self-penalization on the performance of our proposed QUEST method. These variants, denoted as $\mathcal{L}_{\text{SIC}}$, $\mathcal{L}_{\text{UIC}}$, $\mathcal{L}_{\text{P-UIC}}$, are evaluated through modality-specific decoder objectives in our ablation studies. The evaluation is conducted on the ICR task, comparing the performance differences between CLIP and VSE++ models on the Flickr30k and MS-COCO datasets.

**Decoder Configuration and Objective Function Assignment.** As shown in Table 2, both $\mathcal{L}_{\text{SIC}}$ and $\mathcal{L}_{\text{UIC}}$ individually exhibit performance drops compared to the full QUEST. CLIP and VSE++ perform poorly when only using unique information constraints, achieving 55.1 and 40.7 on COCO and 80.8 and 47.8 on Flickr30k, respectively, suggesting that capturing unique information alone is insufficient for improvement. Furthermore, the results demonstrate that self-penalization enhances the performance of QUEST, preventing overconfidence and retaining more potential positive samples. CLIP and VSE++ have better performance when using $\mathcal{L}_{\text{P-UIC}}$ instead of $\mathcal{L}_{\text{UIC}}$. In addition, our QUEST method outperforms the approach utilizing $\mathcal{L}_{\text{SIC+UIC}}$. The ablation study aligns with our theoretical framework, confirming that simultaneously capturing shared and unique information in multi-view contrastive learning, along with the application of self-penalization, leads to significant improvements in downstream tasks.

**Unique Decoder Degeneration.** Our architecture includes a unique decoder module. To evaluate its impact on the task, we replaced this unique decoder with a shared decoder to determine if performance improvements were due to parameter changes. As shown in Table 2, using dual shared decoders ($\mathcal{L}_{\text{SIC+SIC}}$) did not enhance performance and even resulted in a decline compared to a single shared decoder ($\mathcal{L}_{\text{SIC}}$). We refer to this decline as the degradation of the unique decoder.

## 3.4 Evaluation on More Modalities

We conducted extensive experiments on three modalities: image, text and audio. For image-audio evaluations, we leveraged the FMA [19, 20] and GTZAN [69] datasets, while text-audio experiments were evaluated on CLOTHO [21] and AUDIOCAPS [36] datasets. As shown in Figure 4, our pro-

Table 2: Ablation study on image caption retrieval task with different training objectives. D1 and D2 denote decoders in the architecture. Decoder with all ✗beneath are omitted, while those with ✓indicate optimization with corresponding objective functions. **Bold** and underlined numbers indicate the best and second-best results, respectively.

| Methods | | | | Flickr30k | | | | | | | MS-COCO | | | | | | |
|---|---|---|---|---|---|---|---|---|---|---|---|---|---|---|---|---|---|
| D1 | D2 | | | *i2t* | | | *t2i* | | | RSUM | *i2t* | | | *t2i* | | | RSUM |
| $\mathcal{L}_{SIC}$ | $\mathcal{L}_{SIC}$ | $\mathcal{L}_{UIC}$ | $\mathcal{L}_{P\text{-}UIC}$ | R@1 | R@5 | R@10 | R@1 | R@5 | R@10 | | R@1 | R@5 | R@10 | R@1 | R@5 | R@10 | |
| | | | | **CLIP** | | | | | | | **CLIP** | | | | | | |
| ✓ | ✗ | ✗ | ✗ | 86.9 | 97.4 | 98.8 | 72.4 | **92.1** | **95.8** | 543.4 | 63.8 | 86.1 | 92.3 | 46.3 | 74.8 | 84.1 | 447.4 |
| ✗ | ✗ | ✓ | ✗ | 80.8 | 94.4 | 96.5 | 66.9 | 88.4 | 92.9 | 519.9 | 55.1 | 81.6 | 89.4 | 43.3 | 72.5 | 82.9 | 424.8 |
| ✗ | ✗ | ✗ | ✓ | 85.5 | 96.6 | 98.1 | 70.0 | 87.3 | 90.7 | 528.2 | 63.0 | 85.9 | 91.7 | 45.2 | 70.4 | 79.2 | 435.4 |
| ✓ | ✗ | ✓ | ✗ | 81.9 | 96.2 | 98.0 | 69.5 | 90.0 | 94.2 | 529.8 | 64.9 | 86.6 | 92.8 | 47.3 | 75.1 | 83.9 | 450.6 |
| ✓ | ✓ | ✗ | ✗ | 76.6 | 92.3 | 95.9 | 59.0 | 84.4 | 90.9 | 499.1 | 54.4 | 79.2 | 86.7 | 39.9 | 68.4 | 78.9 | 407.5 |
| ✓ | ✗ | ✗ | ✓ | **89.3** | **97.8** | **99.2** | **73.9** | 91.5 | 95.0 | **546.7** | **65.4** | **87.7** | **93.6** | **48.5** | **75.7** | **84.7** | **455.6** |
| | | | | **VSE++** | | | | | | | **VSE++** | | | | | | |
| ✓ | ✗ | ✗ | ✗ | 52.6 | 79.8 | 87.8 | 39.5 | 69.8 | 79.4 | 408.9 | 42.2 | 72.7 | 83.2 | 30.9 | 61.2 | 73.5 | 363.7 |
| ✗ | ✗ | ✓ | ✗ | 47.8 | 72.8 | 80.8 | 36.7 | 63.2 | 73.3 | 374.6 | 40.7 | 71.2 | 82.1 | 30.3 | 60.5 | 73.0 | 357.8 |
| ✗ | ✗ | ✗ | ✓ | 49.0 | 74.1 | 81.3 | 36.4 | 64.1 | 73.1 | 378.0 | 40.9 | 71.4 | 82.4 | 30.8 | 60.6 | 73.2 | 359.3 |
| ✓ | ✗ | ✓ | ✗ | 53.3 | 79.8 | 87.6 | 40.5 | 68.1 | 78.0 | 407.3 | 44.9 | 74.1 | 84.4 | 32.3 | 62.8 | 74.7 | 373.2 |
| ✓ | ✗ | ✗ | ✓ | **54.7** | **80.3** | **88.2** | **42.0** | **70.3** | **79.6** | **415.1** | **45.3** | **75.5** | **85.4** | **34.1** | **64.5** | **76.3** | **381.1** |

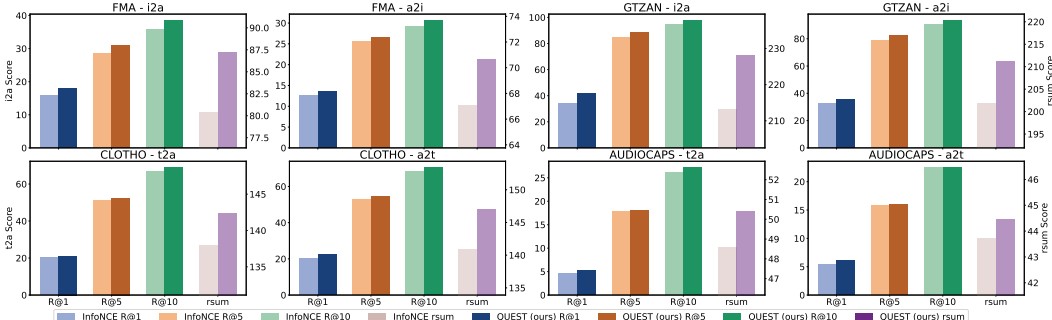

Figure 4: Performance comparison of InfoNCE and QUEST methods with additional audio modality on image-to-audio (i2a) and audio-to-image (a2i) retrieval tasks across FMA, GTZAN, CLOTHO, and AUDIOCAPS datasets.

posed methods outperformed the baseline. For simplicity, we employed the almost simplest decoder structure (linear layers) and did not implement modality fusion or any cross-modal interactions.We believe that enhancing the architecture, strengthening cross-modal interactions, employing a larger batch size and incorporating pre-training would yield even higher performance.

## 3.5   Case Study

On Flickr30k, $\mathcal{L}_{InfoNCE}$ over-privileges shared information, particularly for synthetic shortcut data, while $\mathcal{L}_{QUEST}$ emphasizes task-relevant semantic information. In the text-to-image retrieval case, $\mathcal{L}_{InfoNCE}$ leverages modality-shared information like "gold" and "bicycle" for retrieval. When shared information is explicitly injected across modalities through shortcuts, $\mathcal{L}_{InfoNCE}$ pays less attention to the semantic information in queries.

# 4   Related Work

## 4.1   Multimodal Contrastive Learning

In the field of multimodal contrastive learning, substantial progress has been achieved in harnessing the synergistic effects of multimodal data [38, 46, 11, 74]. Multimodal contrastive learning mainly concentrates on developing methods to semantically align data from diverse modalities through contrastive learning techniques, including optimizing contrastive objectives to enhance agreement between paired data [52, 33, 79, 15], improving the selection of positive and negative sample [85, 35, 64], optimize training mechanism [28, 12] and designing innovative network architectures

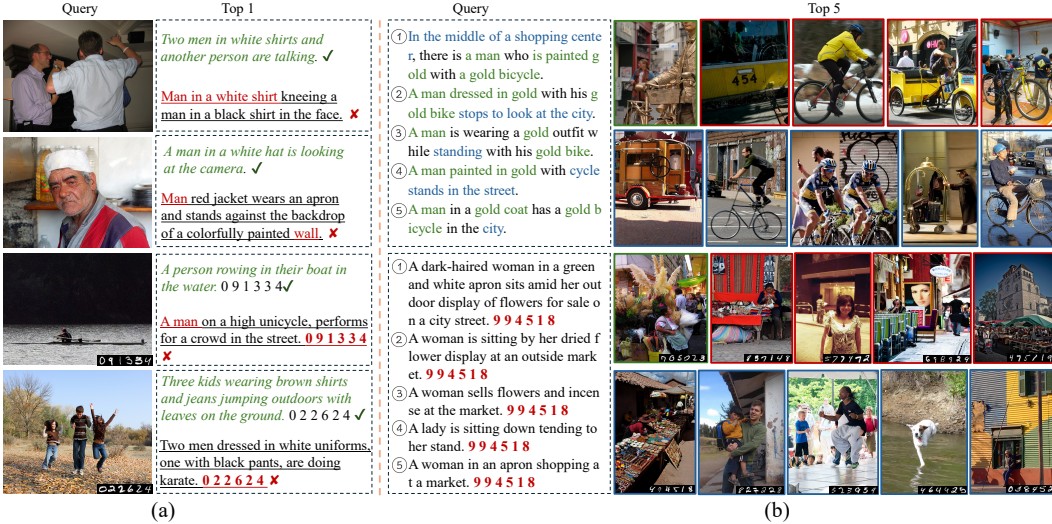

Figure 5: Case Study: (a) Image-to-text retrieval , where the results of $\mathcal{L}_{\text{QUEST}}$ and $\mathcal{L}_{\text{InfoNCE}}$ are denoted by italics and underlines, respectively. (b) Text-to-image retrieval, where red and green borders indicate the top-5 retrievals using $\mathcal{L}_{\text{QUEST}}$, while blue borders represent those using $\mathcal{L}_{\text{InfoNCE}}$. The upper and lower sections in both (a) and (b) demonstrate scenarios with and without shortcuts, respectively.

[7, 6, 26, 10]. Nevertheless, our work is orthogonal to most of the aforementioned studies. Our shared branch approach can be integrated with existing training methods such as MoCo [28] and SimCLR [7].

## 4.2 Shortcuts Learning

Shortcut Learning refers to the tendency of deep neural networks to exploit simple but potentially unreliable features (i.e., "shortcuts") in data for decision-making, rather than learning more complex but reliable features [25]. This phenomenon can lead to poor model performance on out-of-distribution data and is particularly common in multimodal retrieval [3] and VQA tasks [55, 18]. Robinson et al. [58] proposed strategically adjusting the feature distribution of positive and negative sample pairs to achieve implicit feature modification in contrastive learning, guiding models to learn more robust feature representations. Sanchez et al. [59] propose maximizing mutual information to capture data attributes in shared and exclusive representations, while minimizing it between them to enforce disentanglement. LTD [4] introduces an additional decoder to reconstruct input text descriptions in the latent space of a universal sentence encoder, preventing image and text encoders from suppressing predictive features. More recently, some works strive to enhance the estimation of mutual information through the utilization of stricter bounds [30, 41, 13, 51] or the introduction of regularization constraints [45], consequently preserving unique information more effectively.

## 5 Conclusion

We introduce QUEST, a framework utilizing specialized decoders to extract both unique and shared information via shared information-guided constraints and self-penalization. This study addresses the challenges of imbalanced negative samples and task-related unique feature suppression in Multimodal Contrastive Learning. Our method optimizes shared and unique representations simultaneously, outperforming state-of-the-art methods in preserving unique information and enhancing contrastive learning. Unlike traditional approaches that employ direct dot products to minimize distances between positive samples, QUEST leverages quaternions for the indirect optimization of unique and shared information. However, the application of cross products in high-dimensional spaces is limited, complicating the control of high-dimensional representations and reducing theoretical interpretability. For further discussion on these limitations, see the appendix (Appendix E).

## Acknowledgements

This work was supported by the grants from the Natural Science Foundation of China (62202029), and Young Elite Scientists Sponsorship Program by CAST (No. 2023QNRC001). Thanks for the computing infrastructure provided by Beijing Advanced Innovation Center for Big Data and Brain Computing. This work was also sponsored by CAAI-Huawei MindSpore Open Fund. Haoyi Zhou is the corresponding author.

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

# Appendix

## A    Broder Impact

Multimodal contrastive learning, as an emerging learning paradigm, enhances the model's performance in understanding and processing multimodal data by establishing connections and comparisons between multiple modalities. With the increasing demand for processing multimodal data, widespread application of large models, and rapid development of embodied intelligence, the application of multimodal contrastive learning in the real world requires more consideration of safety and application.

**Safety** Privacy and data security [5] have exacerbated concerns in the context of sensitive data processing, such as facial recognition and personal identification information in multimodal learning. We adopted open-source datasets to avoid incorporating private information into the training process as much as possible. Concurrently, the training of multimodal algorithms often exhibits biases [48, 27, 14] that can contribute to discrimination or unfair treatment of certain groups, reducing the fairness and inclusivity. Our approach may require bias detection or the application of bias reduction methods [71] in real-world applications to ensure that multimodal algorithms maintain unbiased ethics.

**Application** Multimodal learning, particularly when integrated into embodied intelligence, has far-reaching impacts across various domains. In healthcare [22, 61], it enhances advanced diagnostic tools and personalized treatments, which also requires specific measures to assure patient privacy and data security. In education [81, 31], multimodal AI can personalize learning experiences, potentially reducing educational disparities, which requires an unbiased AI. Embodied intelligence [53] in social interaction facilitates accessibility and provides companionship for vulnerable populations, especially as pre-training then fine-tuning becomes the mainstream training paradigm, our method helps models learn more modal-independent unique information; unlike models such as CLIP, which focus too much on shared information.

**Future work** Our research mainly focuses on extracting unique information in modalities from different views. In terms of model architecture, we have implemented a unique decoder to extract unique information and train the model with the quadruple loss with constraints. Our approach and Bleeker [3] et.al both adopt the supervised learning method. Future work could explore extracting unique information through self-supervised learning (SSL) approaches. One potential direction for this exploration involves augmenting the input data. Liang et al. [41] proposed FactorCL to factorize information into shared information and unique information by augmentation to implement SSL. Furthermore, in the era of large language models (LLM), leveraging the powerful capabilities of LLMs to explicitly extract existing data to unique and shared information between different views, as shown in Fig 6. By combining this with the original data, training can be conducted on this explicitly separated information for enhanced model performance.

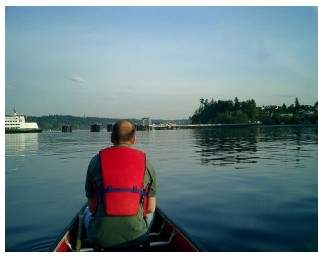 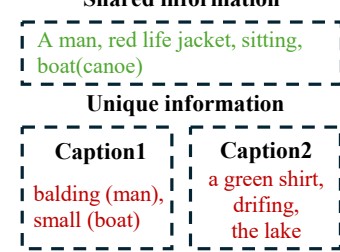

Figure 6: Shared and unique Information in Multimodal Multi-view Scenario. A single image can be described from multiple viewpoints, each containing shared information and distinctive details unique to the specific perspective.

# B    Experiment Details

## B.1    Datasets

**Flickr30k** is a benchmark commonly used in computer vision (CV) and natural language processing (NLP), The tasks applicable to this dataset involving image-caption and multimodal learning, like image understanding, visual question answering and generating text descriptions for images. Flickr30k contains 31783 images collected from the Flickr platform, each with five corresponding descriptive captions, totalling 158,915 captions.

**Microsoft Common Objects in Context (MS-COCO)** is a large-scale dataset containing over 328000 images, and each image is paired with at least five detailed captions annotated by humans for other CV tasks like segmentation; MS-COCO also provides segmentation masks, key points and relationships between objects. The images within the MS-COCO cover a multitude of object categories, activities, and scenes, representing a broad spectrum of settings and contexts.

**Free Music Archive (FMA)** is an extensive, open-access dataset designed for Music Information Retrieval (MIR) research, encompassing 917 GiB of audio data (equivalent to 343 days of playback) from 106,574 Creative Commons-licensed tracks. This diverse collection, spanning 16,341 artists, 14,854 albums, and 161 hierarchically organized genres, provides researchers with high-quality, full-length audio files, pre-computed features, and rich metadata including track and user information, tags, and textual descriptions. FMA's comprehensive nature makes it ideal for various MIR tasks such as genre classification, artist identification, and music recommendation, while its predefined train/validation/test splits and subsets of varying sizes facilitate reproducible research and benchmarking in the field. Code, data, and usage examples are available at `https://github.com/mdeff/fma`.

**GTZAN** is a benchmark dataset widely used in Music Information Retrieval (MIR) and audio signal processing research, particularly for tasks involving musical genre classification and audio feature extraction. The dataset comprises 1,000 audio excerpts, each 30 seconds in duration, equally distributed across 10 distinct musical genres: blues, classical, country, disco, hip-hop, jazz, metal, pop, reggae, and rock. All audio samples in GTZAN are standardized to 22,050 Hz sampling rate, mono channel, and 16-bit resolution in WAV format, facilitating consistent analysis and algorithm development. Despite some noted limitations, GTZAN remains a valuable resource for evaluating and comparing various approaches in automatic music genre recognition and related MIR tasks.

**Clotho** is a diverse audio captioning dataset comprising 4981 audio samples (15-30 seconds each) from Freesound, paired with 24,905 crowdsourced captions (8-20 words each). Designed to facilitate general audio content description using free text, Clotho emphasizes perceptual diversity by providing multiple captions per audio and excluding visual or contextual cues during annotation. The dataset's post-processing, including removal of unique words and speech transcription, enhances its suitability for developing and evaluating audio captioning systems.

**AudioCaps** is a seminal dataset for audio captioning, comprising 46,000 audio clips from AudioSet with human-authored descriptions. This large-scale corpus has become the benchmark for evaluating audio captioning models, catalyzing advancements in audio representation and multimodal learning. AudioCaps has facilitated the development of innovative architectures such as the Audio Captioning Transformer and retrieval-augmented models, significantly contributing to audio-language research. Its impact extends beyond captioning, influencing broader studies in audio-visual integration and inspiring more comprehensive datasets in the field.

## B.2    Experimental Settings

Table 3: Multimodal Model Training Details.

| Model | Flickr30k | | | | | | MS-COCO | | | | | | Visual Encoder | Text Encoder | Params |
|---|---|---|---|---|---|---|---|---|---|---|---|---|---|---|---|
| | Epoch | BS | optimizer | lr | warmup_steps | lr_scheduler | Epoch | BS | optimizer | lr | warmup_steps | lr_scheduler | | | |
| VSE++ | 30 | 128 | adam | 2e-4 | 0 | stepLR | 30 | 128 | adam | 2e-4 | 0 | stepLR | Resnet152 | GRU | 67M |
| CLIP | 5 | 256 | adamw | 2e-5 | 100 | cosine_annealing | 5 | 256 | adamw | 2e-5 | 100 | cosine_annealing | ViT-B/32 | Transformer | 152M |

**Detailed training settings** In our experiments, we conducted a comprehensive comparison of multimodal model training strategies using the Flickr30k and MS-COCO datasets. We used VSE++ and CLIP, each employing distinct training configurations tailored to their respective architectures and optimization requirements.

For the VSE++ model, we utilized a training regime consisting of 30 epochs with a batch size of 128. The optimization process was driven by the Adam optimizer with a learning rate set at $2 \times 10^{-4}$. Notably, no warmup steps were employed, and the learning rate was adjusted using a stepLR scheduler. This configuration was consistently applied across both the Flickr30k and MS-COCO datasets. The visual encoder for VSE++ was based on ResNet152, while the text encoding was handled by a GRU, resulting in a total parameter count of 67 million.

Moreover, the CLIP model was trained over five epochs with a larger batch size of 256. Optimization was performed using the AdamW optimizer with a significantly lower learning rate of $2 \times 10^{-5}$. In contrast to VSE++, CLIP incorporated 100 warmup steps and utilized a cosine annealing scheduler for learning rate adjustment. This setup was also uniformly applied to both datasets. The visual encoder for CLIP was a ViT-B/32, and text encoding was managed by a Transformer, leading to a substantially larger parameter count of 152 million.

**Unique Decoder Implementation Details** We used CLIP(ViT/B-32 backbone) and VSE++(resnet152 backbone); we chose a single-layer MLP for vse++ and two-layer Transformers for CLIP. Both of them use nine layers as a unique start layer and 0 for no unique start layer. We choose the hyperparameters $alpha\_t$ as 0.08 on most experiments and set $positive\_sample$ to false.

## B.3    More Implementation Details

The $\mathcal{L}_{\text{UIC}}$ algorithm enhances contrastive loss in representation learning through a novel embedding space involving quadruplets. This approach primarily comprises two stages: the computation of a similarity map from quadruple embeddings and the subsequent calculation of cross-entropy loss based on this map. The method leverages shared and unique embeddings to construct expressive feature representations, further augmented by three-dimensional vectors generated via cross-products, thereby intensifying the discriminative power of embeddings.

As shown in 1, the algorithm firstly processes four embedding vectors:$x_{\text{shared}}$, $x_{\text{unique}}$, $y_{\text{shared}}$, and $y_{\text{unique}}$, representing shared and unique features across two data sets. Padding may be applied to ensure dimensional consistency (Divisible by 3) across embeddings. The GETSIMMAP function, shown in 2 then calculates the similarity map between $x_{\text{shared}}$ and $y_{\text{shared}}$, reflecting the similarity between shared feature embeddings. This computation involves normalizing the embeddings, obtaining a preliminary similarity matrix via dot products, and adjusting the similarity values through exponential weighting and diagonal normalization.

Upon obtaining the similarity map, the algorithm transforms $x_{\text{shared}}$ and $x_{\text{unique}}$ (along with their $y$ counterparts) into three-dimensional vectors, or *triplet embeddings*, via cross products. This geometric transformation aims to further enhance the distinctiveness of embeddings. The triplet embeddings are then normalized to ensure numerical stability during dot product calculations. Ultimately, the algorithm calculates the dot product of $x_{\text{triplet}}$ and $y_{\text{triplet}}$, scales the result, and applies element-wise multiplication with the similarity map to produce the final *logits*. These logits, after scaling and absolute value adjustments, are used to compute the cross-entropy loss.

We also use orthogonal cosine embedding loss 3 to quantify the similarity between pairs of input vectors, facilitating the training of models on tasks that distinguish between similar and dissimilar data points. The objective of this loss computation is to minimize the discrepancy between predicted logits and actual labels, thereby optimizing the representational capacity of the embeddings.

The synthetic shortcuts experiment enables a quantitative analysis of the model's reliance on shortcuts when they are present and its ability to capture shared and task-relevant unique information. The results demonstrate that our proposed method effectively mitigates the feature suppression phenomenon, contributing to improved performance on downstream tasks compared to previous approaches.

**Generalization Capability Across Multiple Modalities** To evaluate the generalization capability of our proposed approach on more modalities, we conducted comprehensive experiments across various modalities, including visual, textual, and acoustic domains. Specifically, we performed image-audio retrieval experiments utilizing the FMA (Free Music Archive) and GTZAN datasets, with quantitative results presented in Table 4. For text-audio retrieval evaluations, we leveraged the CLOTHO and AUDIOCAPS datasets, with comparative performance metrics detailed in Table 5.

The empirical results demonstrate that our proposed models consistently outperformed the baseline approaches across all experimental configurations. For simplicity, we employed the almost simplest

**Algorithm 1** $\mathcal{L}_{\text{UIC}}$ loss calculation

---

**Require:** Latent representation $x_{\text{shared}}, x_{\text{unique}}, y_{\text{shared}}, y_{\text{unique}}$.
**Ensure:** $\mathcal{L}_{\text{UIC}}$
 1: **function** CALCULATE $\mathcal{L}_{\text{UIC}}(x_{\text{shared}}, x_{\text{unique}}, y_{\text{shared}}, y_{\text{unique}})$
 2:  $B, C \leftarrow$ shape of $x_{shared}$
 3:  **if** padding is enabled in config **then**
 4:    $pad\_size \leftarrow (3 - C \mod 3) \mod 3$
 5:    $x_{shared} \leftarrow$ pad $x_{shared}$ with $pad\_size$
 6:    $x_{unique} \leftarrow$ pad $x_{unique}$ with $pad\_size$
 7:    $y_{shared} \leftarrow$ pad $y_{shared}$ with $pad\_size$
 8:    $y_{unique} \leftarrow$ pad $y_{unique}$ with $pad\_size$
 9:  **end if**
10:  $sim\_map \leftarrow$ GETSIMMAP$(x_{shared}, y_{shared})$
11:  $mini\_heads \leftarrow$ integer division of $C$ by 3
12:  $participated\_dims \leftarrow mini\_heads \times 3$
13:  $x_{shared} \leftarrow$ reshape $x_{shared}[:, : participated\_dims]$ to $(B, -1, 3)$
14:  $x_{unique} \leftarrow$ reshape $x_{unique}[:, : participated\_dims]$ to $(B, -1, 3)$
15:  $y_{shared} \leftarrow$ reshape $y_{shared}[:, : participated\_dims]$ to $(B, -1, 3)$
16:  $y_{unique} \leftarrow$ reshape $y_{unique}[:, : participated\_dims]$ to $(B, -1, 3)$
17:  $x_{uic} \leftarrow$ cross product of $x_{shared}$ and $x_{unique}$ along dimension 2, then reshape to $(B, -1)$
18:  $y_{uic} \leftarrow$ cross product of $y_{shared}$ and $y_{unique}$ along dimension 2, then reshape to $(B, -1)$
19:  $x_{uic} \leftarrow$ normalize $x_{uic}$ along the last dimension
20:  $y_{uic} \leftarrow$ normalize $y_{uic}$ along the last dimension
21:  $logits \leftarrow x_{uic} \times y_{uic}^T$
22:  $logits \leftarrow$ absolute value of $logits$
23:  $logits \leftarrow scale \times logits \times sim\_map$
24:  $num\_logits \leftarrow B$
25:  $labels \leftarrow$ range from 0 to $num\_logits - 1$
26:  $\mathcal{L}_{\text{QUAD}} \leftarrow$ CE loss of $logits$ with $labels$ + CE loss of $logits^T$ with $labels$
27:  $\mathcal{L}_{\cos} \leftarrow$ GETCOSLOSS$(y_{shared}, y_{unique})$ + GETCOSLOSS$(x_{shared}, x_{unique})$
28:  $\mathcal{L}_{\text{UIC}} \leftarrow \mathcal{L}_{\text{QUAD}} + \mathcal{L}_{\cos}$
29:  **return** $\mathcal{L}_{\text{UIC}}$
30: **end function**

---

**Algorithm 2** Calculate similarity map

---

**Require:** latent representations $x$, $y$
**Ensure:** similarity map $sim\_map$
 1: **function** GETSIMMAP$(x, y)$
 2:  $x \leftarrow$ normalize $x$ along the last dimension
 3:  $y \leftarrow$ normalize $y$ along the last dimension
 4:  $sim\_map \leftarrow x \times y^T$
 5:  $sim\_map \leftarrow$ clamp $sim\_map$ between 0 and 1
 6:  $sim\_map \leftarrow \exp(sim\_map)$
 7:  fill the diagonal of $sim\_map$ with 1
 8:  **return** $sim\_map$
 9: **end function**

---

**Algorithm 3** $\mathcal{L}_{\text{cos}}$ loss calculation

**Require:** unique_information $x_{\text{unique}}$, shared_information $x_{\text{shared}}$
**Ensure:** $\mathcal{L}_{\text{cos}}$
 1: **function** GETCOSLOSS($x_{\text{unique}}, x_{\text{shared}}$)
 2:     $\epsilon \leftarrow 1e - 12$
 3:     $product\_sum \leftarrow (x_{\text{unique}}, x_{\text{shared}}).\text{sum}(dim = 1)$
 4:     $matnitude\_square_1 \leftarrow (x_{\text{unique}}, x_{\text{unique}}).\text{sum}(dim = 1) + \epsilon$
 5:     $matnitude\_square_2 \leftarrow (x_{\text{shared}}, x_{\text{shared}}).\text{sum}(dim = 1) + \epsilon$
 6:     $denominator \leftarrow \sqrt{mag\_square1 \times mag\_square2}$
 7:     $cos \leftarrow product\_sum/denominator$
 8:     $zeros \leftarrow$ tensor of zeros with same shape as $x_{\text{unique}}$ along dimension 0
 9:     $pos\_loss \leftarrow 1 - cos$
10:     $neg\_loss \leftarrow \text{clamp}(|cos - 0.0|, \min = 0)$
11:     $target \leftarrow$ tensor of $-1$ with same shape as $x_{\text{unique}}$ along dimension 0
12:     Initialize $loss\_pos$ and $loss\_neg$ with the same shape as $target$, $pos\_loss$, and $zeros$
13:     **for** each index $i$ in $target$ **do**
14:         **if** $target[i]$ is 1 **then**
15:             $loss\_pos[i] \leftarrow pos\_loss[i]$
16:         **else**
17:             $loss\_pos[i] \leftarrow zeros[i]$
18:         **end if**
19:         **if** $target[i]$ is -1 **then**
20:             $loss\_neg[i] \leftarrow neg\_loss[i]$
21:         **else**
22:             $loss\_neg[i] \leftarrow zeros[i]$
23:         **end if**
24:     **end for**
25:     $loss \leftarrow loss\_pos + loss\_neg$
26:     **return** mean of $loss$
27: **end function**

Table 4: Image audio retrieval results on FMA and GTZAN datasets.

| Method | Datasets | i2a | | | | a2i | | | |
|---|---|---|---|---|---|---|---|---|---|
| | | R@1 | R@5 | R@10 | RSUM | R@1 | R@5 | R@10 | RSUM |
| InfoNCE | FMA | 15.87 | 28.62 | 35.87 | 80.36 | 12.50 | 25.50 | 29.12 | 67.12 |
| QUEST | | **17.83** | **30.87** | **38.50** | **87.20** | **13.50** | **26.52** | **30.62** | **70.64** |
| InfoNCE | GTZAN | 34.01 | 84.73 | 94.41 | 213.15 | 32.48 | 78.68 | 90.86 | 202.02 |
| QUEST | | **41.62** | **88.83** | **97.65** | **228.1** | **35.53** | **82.23** | **93.40** | **211.16** |

decoder structure (linear layers) and did not implement modality fusion or any cross-modal interactions. We believe that enhancing the architecture, strengthening cross-modal interactions, employing a larger batch size and incorporating pre-training would yield even higher performance.

**Compute Resources.** All experiments in this paper are run on a single NVIDIA A100 GPU. The implementation is based on PyTorch 2.0.1. It takes about 2 hours to train the CLIP-based model on Flickr30K for 5 epochs, and the maximum training time for other experiments does not exceed 12 GPU hours.

Table 5: Text audio retrieval results on CLOTHO and AUDIOCAPS datasets.

| Method | Datasets | t2a | | | | a2t | | | |
|---|---|---|---|---|---|---|---|---|---|
| | | R@1 | R@5 | R@10 | RSUM | R@1 | R@5 | R@10 | RSUM |
| InfoNCE | CLOTHO | 20.16 | 51.30 | 66.56 | 138.02 | 20.06 | 52.66 | 68.23 | 140.95 |
| QUEST | | **21.10** | **52.45** | **68.86** | **142.41** | **22.36** | **54.23** | **70.42** | **147.01** |
| InfoNCE | AUDIOCAPS | 4.59 | 17.79 | 26.22 | 48.6 | 5.45 | 15.78 | 22.48 | 43.71 |
| QUEST | | **5.16** | **18.08** | **27.17** | **50.41** | **6.02** | **15.98** | **24.78** | **46.78** |

# C  Notation

Table 6: Notation used in the paper.

| Symbol | Description |
| --- | --- |
| $\mathcal{M}_i$ | Modality $i$ |
| $x_i^j$ | The $i$ th sample(view) of modality $j$ |
| $\mathbf{H}_{\mathcal{M}_i}^j$ | Representations as the encoder's output |
| $\mathbf{Z}_i^{\mathbf{s}}, \mathbf{Z}_i^{\mathbf{u}}$ | $\mathbf{Z}_i^{\mathbf{s}}$ and $\mathbf{Z}_i^{\mathbf{u}}$ denote the decoder's output representations for shared and unique information, respectively. |
| $\mathbf{Z}_i^{\mathbf{n}}$ | The cross product of $\mathbf{Z}_i^{\mathbf{s}}$ and $\mathbf{Z}_i^{\mathbf{u}}$ |
| $\mathcal{L}_{\text{InfoNCE}}$ | InfoNCE loss |
| $\mathcal{L}_{\text{InfoNCE+LTD}}$ | Loss that combines InfoNCE and LTD |
| $\mathcal{L}_{\text{InfoNCE+IFM}}$ | Loss that combines InfoNCE and IFM |
| $\mathcal{L}_{\text{SIC}}$ | Shared information constraint loss |
| $\mathcal{L}_{\text{UIC}}$ | Unique information constraint loss |
| $\mathcal{L}_{\text{P-UIC}}$ | Penalized $\mathcal{L}_{\text{UIC}}$ |
| $\widetilde{\mathcal{L}}_{\text{P-UIC}}$ | Reformulated $\mathcal{L}_{\text{P-UIC}}$, ignore second item |
| $\mathcal{L}_{\text{cos}}$ | Orthogonalized cosine loss |
| $\mathcal{L}_{\text{QUEST}}$ | Overall Quardruple InfoNCE Loss |
| $\mathbf{S}$ | The similarity matrix between shared embedding |
| $\mathcal{P}$ | The weighted similarity matrix between shared embedding $\mathbf{Z}_i^{\mathbf{s}}$ and $\mathbf{Z}_j^{\mathbf{s}}$ |
| $\mathcal{B}$ | Batch of image-caption pairs |
| $\tau$ | temperature coefficient |
| $X_A$ | random variables from modility $\mathcal{M}_A$ |
| $\mathcal{P}$ | the weighted similarity matrix, utilized as penalization |
| $\mathcal{F}_{\mathcal{M}}(\cdot; \Theta)$ | Modality $\mathcal{M}$ encoder parametrised by $\Theta$; takes sample $X_{\mathcal{M}}^j$ as input and returns its latent representation $\mathbf{H}_{\mathcal{M}}^j = \mathcal{F}_{\mathcal{M}}(x^j; \theta)$ |
| $\mathcal{G}_{\mathcal{M}}(\cdot; \Phi)$ | Modality $\mathcal{M}$ decoder parametrised by $\phi$; takes representations $\mathbf{H}_{\mathcal{M}}^j$ as input and returns its latent representation $\mathbf{Z}_{\mathcal{M}}^j = \mathcal{G}_{\mathcal{M}}(\mathbf{H}_{\mathcal{M}}^j; \Phi)$ |
| $s(\cdot, \cdot)$ | Scoring function |
| $H(Z_j \mid Z_i)$ | The conditional entropy of $Z_j$ given $Z_i$ |
| $H^{\tilde{P}}(Z_j \mid Z_i)$ | Conditional entropy under the penalized term $\tilde{P}$ |
| $I(Z_i, Z_j)$ | Mutual information between $Z_i$ and $Z_j$ |

# D  Analysis.

## D.1  Mutual Information

In many machine learning tasks, it is often observed that different views simultaneously harbour both task-relevant shared information and unique information (e.g., image captioning [38, 40, 72] and referring expression segmentation [78, 44]). To this end, we work with a dual-encoder architecture.

Contrastive learning with multiple views obtains mutual information between different modalities by learning the similarity among different views. [70] et al. introduced a lower bound on mutual information, known as InfoNCE, which is based on the concept of Noise Contrastive Estimation (NCE); it compares the compatibility of different views by maximizing the mutual information of positive pairs and minimizing the mutual information of negative pairs, learning to extract the consistent representation across different modal. It can be defined as follows:

$$I_{\text{InfoNCE}}(X;Y) = \mathbb{E}_{(x,y)\sim p(x,y)} \left[ \frac{1}{N} \sum_{i=1}^{N} \log \frac{\exp f(x_i, y_i)}{\frac{1}{N} \sum_{j=1}^{N} \exp f(x_i, y_j)} \right] \tag{13}$$

In this equation, $x_i$ and $y_i$ represent paired samples from the joint distribution of the two random variables under consideration, while $y_j$ represents samples from the marginal distribution of one of the variables. The function $f(x,y)$ is a learnable function, often parameterized by a neural network, that aims to distinguish between the paired samples and the independently sampled ones.

In the realm of vison-language contrastive learning, the *InfoNCE loss function* [70] enhances the similarity between positive sample pairs relative to negative ones. Mathematically, the InfoNCE loss is articulated as:

$$\mathcal{L}_{\text{InfoNCE}} = -\log \frac{\exp(\text{sim}(x, y^+)/\tau)}{\sum_{i=0}^{K} \exp(\text{sim}(x, y_i)/\tau)} \tag{14}$$

where $\text{sim}(x,y)$ calculates the similarity between samples $x$ and $y$, typically computed through dot product or cosine similarity. $y^+$ denotes the positive sample corresponding to $x$, and $\{y_i\}_{i=0}^{K}$ is a set including one positive and $K$ negative samples. The parameter $\tau$ serves as a temperature coefficient, modulating the scale of similarity scores.

InfoNCE loss prevails in many contrastive learning algorithms and performs well in applications. Multi-view redundancy [66, 2, 30, 63, 65, 67] assumes that there exists duplicated information across varied views or representations. Contrastive losses like InfoNCE loss tend to maximize the mutual features between different views while suppressing task-relevant features that may be used in downstream tasks in multi-view representation learning.

Mutual information (MI), often denoted as $I(X;Y)$, is a fundamental concept in information theory that quantifies the statistical dependence between two random variables, $X$ and $Y$. It measures the reduction in uncertainty about one variable when the value of the other is known and is defined as:

$$I(x;y) = \mathbb{E}_{(x,y)\sim p(x,y)}[\log \frac{p(x,y)}{p(x)p(y)}] \tag{15}$$

where $p(x,y)$ is the joint distribution, and $p(x)$ and $p(y)$ are the marginal distributions of $X$ and $Y$.

In machine learning, particularly in deep learning, MI is often used as an objective function or regularization component to promote or constrain the interdependence among variables. However, the precise quantification of MI is only attainable in limited instances, as it requires the closed form of the density function and the tractable logarithm density ratio between the joint and marginal distributions. In most machine learning applications, practitioners only have access to samples from the joint distribution, making the direct computation of MI infeasible.

The InfoNCE bound has several desirable properties that make it an attractive choice for estimating mutual information. Firstly, it is a lower bound on the true mutual information, $I(X;Y) \geq I_{\text{NCE}}$, which means that maximizing the InfoNCE bound leads to an increase in the true mutual information.

Secondly, the bound is computationally tractable and can be efficiently optimized using standard gradient-based methods, such as stochastic gradient descent.

As shown in Fig 2, multimodal mutual information can be divided into shared information $I(X_A; X_B; Y)$ and task-relevant unique information ($I(x_A; Y|X_B)$ and $I(X_B; Y|X_A)$). Information useful for the task can be represented as

$$
\begin{aligned}
I_\tau &= H(Y) - H(Y|X_A, X_B) \\
&= I(X_A; X_B; Y) + I(X_A; Y|X_B) + I(X_B; Y|X_A)
\end{aligned}
\tag{16}
$$

### D.2  Gradient Analysis

For the simplified loss function Eq. (9), we first calculate the partial derivative of $\mathbf{Z}_i^{\mathbf{s}}$:

$$
\frac{\partial \widetilde{\mathcal{L}}}{\partial \mathbf{Z}_i^{\mathbf{n}}} = \frac{1}{\lambda} \frac{\partial \lambda}{\partial \mathbf{Z}_i^{\mathbf{n}}} - \frac{\mathcal{P}^+}{\tau} \frac{\partial s^+}{\partial \mathbf{Z}_i^{\mathbf{n}}}
\tag{17}
$$

where $s^+ = s(\mathbf{Z}_i^{\mathbf{n}}, \mathbf{Z}_j^{\mathbf{n}^+})$. Define $s^k = s(\mathbf{Z}_i^{\mathbf{n}}, \mathbf{Z}_{jk}^{\mathbf{n}^-})$, then we have:

$$
\begin{aligned}
\frac{\partial \widetilde{\mathcal{L}}}{\partial \mathbf{Z}_i^{\mathbf{n}}} &= \frac{1}{\lambda} \left[ \exp\left(\frac{\mathcal{P}^+ s^+}{\tau}\right) \cdot \frac{\mathcal{P}^+}{\tau} \cdot \frac{\partial s^+}{\partial \mathbf{Z}_i^{\mathbf{n}}} + \sum_{k=1}^m \exp\left(\frac{\mathcal{P}_k s^k}{\tau}\right) \cdot \frac{\mathcal{P}_k}{\tau} \cdot \frac{\partial s^k}{\partial \mathbf{Z}_i^{\mathbf{n}}} \right] - \frac{\mathcal{P}^+}{\tau} \frac{\partial s^+}{\partial \mathbf{Z}_i^{\mathbf{n}}} \\
&= -\frac{\mathcal{P}^+}{\tau} \left(1 - \frac{\exp\left(\frac{\mathcal{P}^+ s^+}{\tau}\right)}{\lambda}\right) \frac{\partial s^+}{\partial \mathbf{Z}_i^{\mathbf{n}}} + \frac{1}{\lambda \tau} \sum_{k=1}^m \mathcal{P}_k \exp\left(\frac{\mathcal{P}_k s^k}{\tau}\right) \frac{\partial s^k}{\partial \mathbf{Z}_i^{\mathbf{n}}} \\
&= -\frac{\mathcal{P}^+}{\tau} \frac{\partial s^+}{\partial \mathbf{Z}_i^{\mathbf{n}}} + \frac{1}{\lambda \tau} \sum_{k=0}^m \mathcal{P}_k \exp\left(\frac{\mathcal{P}_k s^k}{\tau}\right) \frac{\partial s^k}{\partial \mathbf{Z}_i^{\mathbf{n}}}
\end{aligned}
\tag{18}
$$

where $\lambda = \sum_{k=0}^m \exp(\mathcal{P} \cdot s(\mathbf{Z}_i^n, \mathbf{Z}_{jk}^n)/\tau)$. For energy function $s(\mathbf{Z}_i, \mathbf{Z}_j) = \hat{\mathbf{Z}}_i \cdot \hat{\mathbf{Z}}_j = \mathbf{Z}_i \cdot \mathbf{Z}_j/(\|\mathbf{Z}_i\| \|\mathbf{Z}_j\|)$, the derived gradient on the vector $\mathbf{Z}_i$ is shown as:

$$
\begin{aligned}
\frac{\partial s(\mathbf{Z}_i, \mathbf{Z}_j)}{\partial \mathbf{Z}_i} &= \frac{1}{\|\mathbf{Z}_i\|\|\mathbf{Z}_j\|} \left( \mathbf{Z}_j - \frac{\mathbf{Z}_i, \mathbf{Z}_j}{(\|\mathbf{Z}_i\|)^2} \mathbf{Z}_i \right) \\
&= \frac{\mathbf{Z}_j}{\|\mathbf{Z}_i\|\|\mathbf{Z}_j\|} - s(\mathbf{Z}_i \mathbf{Z}_j) \frac{\mathbf{Z}_i}{\|\mathbf{Z}_i\|^2}. \\
&= \frac{1}{\|\mathbf{Z}_i\|} (\hat{\mathbf{Z}}_j - s(\mathbf{Z}_i, \mathbf{Z}_j) \hat{\mathbf{Z}}_i).
\end{aligned}
\tag{19}
$$

### D.3  Mutual Information Estimation

Let $Z_i$ and $Z_i$ represent the compact vector representation from different modalities input; the classical mutual information between $Z_i$ and $Z_j$ can be defined as:

$$
I(Z_i, Z_j) = \sum_{Z_i, Z_j} p(Z_i, Z_j) \log \frac{p(Z_i, Z_j)}{p(Z_i) p(Z_j)}
\tag{20}
$$

According to [70], the density ratio $\frac{p(Z_i, Z_j)}{p(Z_i) p(Z_j)}$ is expressed as $\exp(s(Z_i, Z_j)/\tau)$, where $s(Z_i, Z_j)$ is the similarity score and $\tau$ is the temperature parameter. The penalty term $\mathcal{P}$ is defined as follows:

$$
\mathcal{P} = \exp(\lambda[\mathbf{Z}_i^{\mathbf{s}} \mathbf{Z}_j^T - \text{diag}(\mathbf{Z}_i^{\mathbf{s}} \mathbf{Z}_j^T) + \mathbf{I}]).
\tag{21}
$$

The penalty term does not affect the numerator but amplifies the denominator in Eq. (8) thus the energy function in Eq. (8) can be reformulated as:

$$
\exp(s(Z_i, Z_j)/\tau) \cdot \frac{1}{\tilde{\mathcal{P}}} = \exp(s(Z_i, Z_j)/\tau - \log \tilde{\mathcal{P}}) \propto \frac{\tilde{p}(Z_j|Z_i)}{p(Z_j)} \quad \text{where} \quad \tilde{P} \propto \mathcal{P}
\tag{22}
$$

Next, we derive the lower bound of MI,

$$
\begin{aligned}
\widetilde{\mathcal{L}}_{\text{P-UIC}} &= -\mathbb{E}\log\left[\frac{\frac{\tilde{p}(Z_{jk}|Z_{ik})}{p(Z_{jk})}}{\frac{\tilde{p}(Z_{jk}|Z_{ik})}{p(Z_{jk})} + \sum_{t\neq k}^{N}\frac{\tilde{p}(Z_{jt}|Z_{ik})}{p(Z_{jt})}}\right] \\
&= \mathbb{E}\log\left[1 + \frac{p(Z_{jk})}{\tilde{p}(Z_{jk}|Z_{ik})}\sum_{t\neq k}^{N}\frac{\tilde{p}(Z_{jt}|Z_{it})}{p(Z_{jt})}\right] \\
&\approx \mathbb{E}\log\left[1 + \frac{p(Z_{jk})}{\tilde{p}(Z_{jk}|Z_{ik})}(N-1)\mathbb{E}_{Z_{jt}}\frac{\tilde{p}(Z_{jt}|Z_{it})}{p(Z_{jt})}\right]. \qquad (23)\\
&= \mathbb{E}\log\left[1 + \frac{p(Z_{jk})}{\tilde{p}(Z_{jk}|Z_{ik})}(N-1)\right] \\
&\geq \mathbb{E}\log\left[\frac{p(Z_{jk})}{\tilde{p}(Z_{jk}|Z_{ik})}N\right] \\
&= H^{\tilde{\mathcal{P}}}(Z_j|Z_i) - H(Z_j) + \log N \\
&= H^{\tilde{\mathcal{P}}}(Z_j|Z_i) - I(Z_i, Z_j) - H(Z_j|Z_i) + \log N
\end{aligned}
$$

# E  Limitation

**Dimension restrictions.** The limitations of the cross-product in the context of QupleInfoNCE's unique information extraction are theoretically constrained. When contrastive learning is applied to both unique and shared representations, it can lead to model degradation, and a trade-off choice is the cross-product, which is a weaker constraint. In low-dimensional space, it is straightforward to prove the properties of orthogonal vector space, yet the extension of the cross-product to high-dimensional space is challenging. The behaviour of high-dimensional space representation is difficult to control, resulting in our theory lacking sufficient interpretability in high-dimensional space. This highlights the importance of developing more sophisticated methods for high-dimensional data analysis in the future.

**The Extraction of Unique Information.** On the synthetic shortcuts dataset, our framework achieved significant performance, validating the positive role of unique information in preventing model shortcuts. However, within the entire framework, the core lies in identifying unique information, which is essentially a task-dependent definition. Our experiment indicates that pre-trained models (such as CLIP [52]) typically yield higher unique information benefits due to their extensive generic representations. Non-pre-trained models (such as VSE++ [23]) struggle to discern unique information, tending to shortcuts and losing vital unique information. In this study, we introduced an additional unique information decoder to capture unique information, which incorporated an extra gradient branch, suggesting exploration space remains for single-stream unique information extraction. This also proposes future research: how to more effectively extract unique information from a single stream to reduce computational complexity while maintaining model performance.

