# OpenReview forum: "QUEST: Quadruple Multimodal Contrastive Learning with Constraints and Self-Penalization"
_NeurIPS.cc/2024/Conference — NeurIPS 2024 poster_

### Official Review · Reviewer_zJ9C · 2024-07-09

**Soundness:** 3
**Presentation:** 3
**Contribution:** 3
**Rating:** 8
**Confidence:** 4

**Summary:**

This paper proposes the Quadruple Multimodal Contrastive Learning Framework (QUEST) to capture shared and unique task-relevant information in the process of contrastive learning, enabling models to capture more unique information in downstream tasks to achieve better performance. Specifically, this paper introduces a quadruple embedding space and optimizes shared and unique information within it simultaneously. Additionally, this paper adopts a self-penalization mechanism using shared information to guide the optimization of unique information. This paper evaluates the QUEST method on popular datasets (Flickr30k and COCO). On public benchmarks and synthetic datasets, the method shows significant performance improvements.

**Strengths:**

1. Compared to the latest state-of-the-art methods, a significant performance improvement was achieved (an average of 97.95% on the CLIP model), with enhancements observed across multiple mainstream datasets (flickr30k and coco) and models (both CNN-based model and Transformer-based model).

2. The idea of decomposing features into task-relevant unique features and task-relevant shared features, leveraging quaternion vector spaces, is very novel and effective according to the ablation study. Great work!

3.  The motivation of the paper is very logical, clear, and easy to understand.

4.  The paper demonstrates very high performance on both popular public datasets and the shortcuts dataset.

**Weaknesses:**

1. In the shared Decoder of Equation 1 in Section 2.3, the parameters on both sides of the shared decoder equation are inconsistent: the left side has $\Phi_i$, while the right side has $\Psi_i$. This is strange. Can you tell me why it needs to be written like this if necessary?

2. In the 3.3 ablation study section, what does QUEST-SIC and QUEST-UIC mean? There was no explanation before. Do QUEST-SIC and QUEST-UIC correspond to $\mathcal{L}_{\text{SIC}}$ and $\mathcal{L}_{\text{UIC}} in Table 2 respectively?

3. What does $Z_j^n+$ in equation 5 mean? Can you explain why the "+" sign is placed to the right of the symbol "Z_j^n"?

**Questions:**

How can features be decomposed into shared features and unique features? What is the effectiveness of this method in practice?

**Limitations:**

The behavior of the cross product operation in low dimensions may be unpredictable in higher dimensions.

---

> ### Author Rebuttal · Authors · 2024-08-06
>
> Dear Reviewer zJ9C,
>
> We sincerely thank you for taking the time to review our paper. Our responses to your comments are provided here:
>
> ---
>
> **W1: "In the shared Decoder of Equation 1."**
>
> **A1:** Thank you for your valuable review comments. There is indeed a typo in Equation 1. For shared information, the input data $\mathbf X_i$ is encoded by an encoder parameterized by $\Theta_i$, denoted as $\mathcal F_{\mathcal M_i}$, and then further processed by a shared decoder parameterized by $\Psi_i$, denoted as $\mathcal G^{s}_{\mathcal M_i}$, to obtain the representation $\mathbf Z ^s_i$.
>
> **W2: "What does QUEST-SIC and QUEST-UIC mean?"**
>
> **A2:** In the section of ablation study, QUEST-SIC means only use shared information constraint for training, similarly, QUEST-UIC refers to only using unique information constraint during training. In order to be consistent with the experimental table, QUEST-SIC and QUEST-UIC will be called $\mathcal L_{\text{SIC}}$ and $\mathcal L_{\text{UIC}}$ respectively.
>
> **W3: "What does $Z_j^n +$ in equation 5 mean? Can you explain why the "+" sign is placed to the right of the symbol "$Z_j^n$"?"**
>
> **A3:** Thank you for your correction. Sorry for the small mistake in writing the latex formula. The "n" and "+" should be placed together on the upper right corner of "z", that is, it should be written as $Z_j^{n+}$. This symbol represents the representation of a positive sample in the quaternion embedding space.
>
> **Q1: "How can features be decomposed into shared features and unique features? What is the effectiveness of this method in practice?"**
>
> **A4:** Please see reviewer 9n5E A1. For pre-trained models, it is imperative to retain as much potentially beneficial information for downstream tasks as possible. However, achieving this through contrastive learning in multi-view scenarios presents significant challenges, particularly in cases of many-to-many relationships between images and text. For instance, a single image can often be described by multiple captions, all of which serve as positive samples for that image. Traditional contrastive learning methods tend to prioritize the shared information among captions (as discussed in Section 2.3), potentially leading to the loss of unique information in the encoder. This information loss can be detrimental to downstream tasks. Moreover, the unique information contained in each caption is of  importance. We have conducted extensive experiments on MSCOCO and Flickr30k datasets to substantiate this claim.
>
> **L1: "The behavior of the cross product operation in higher dimensions."**
>
> **A5:** You've raised a valuable point. The generalization of the product can be achieved through multiple approaches, leveraging both the orientation and metric structure analogous to the conventional three-dimensional cross product. In an $n$-dimensional space, it is feasible to compute the product of $n - 1$ vectors, resulting in a vector orthogonal to all input vectors. However, when constrained to non-trivial binary operations yielding vector outputs, the existence of such a product is limited to spaces of three and seven dimensions. Empirically, the properties of high-dimensional cross products may be different from those in low dimensions, so we are discussing the cross product in a limited finite embedding space. Although we have experimentally validated the effectiveness of this method, we are also eager to provide an explanatory approach. For each pair $(A_i, B_i)$, we calculate the cross product in 3-dimensional space: $A_i \times B_i = \mathbf{K}_i A_i B_i$, where $\mathbf{K}_i$ is a zero-diagonal matrix. When extending to higher dimensions, $\mathbf{K}$ can be considered as a sparse projection matrix with fixed parameters.

---

> > ### Comment · Reviewer_zJ9C · 2024-08-10
> >
> > Thanks for the authors' responses, which are clear and comprehensive. I recommend the authors to modify the Eq.5 and add the explanation of w2 to the camera ready version.

---

> > > ### Author Response · Authors · 2024-08-14
> > >
> > > Dear Reviewer zJ9C,
> > >
> > > Thank you very much for your valuable feedback. We will address the corresponding issues in the subsequent versions.

---

### Official Review · Reviewer_9n5E · 2024-07-12

**Soundness:** 4
**Presentation:** 4
**Contribution:** 3
**Rating:** 7
**Confidence:** 4

**Summary:**

The paper focuses on developing a new multi-modal representation learning approach where the extraction and integration of both shared and unique information across multimodal data is the focus. The method aims to pull shared representations closer while aligning the unique representations with the shared representation on a common plane.

Key components: an encoder, a shared decoder, and a unique decoder. Contrastive loss constrains learning of shared information. The proposed framework seeks to mitigate shortcut learning.

**Strengths:**

The proposed idea is novel and makes sense.

Technical details are sufficient to understand the main idea of the paper.

The proposed design and the architecture makes sense.

**Weaknesses:**

Lack of theoretical analysis.

Is there any theoretical justification/demonstration or proof that the encoder information can be disentangled to shared and unique representations for a given distribution w.r.t a task or in a task-agnostic manner? I would like to hear author perspective.

**Questions:**

What does it mean by shortcut learning in this case? There can be large variations in learned shortcuts.

Better to discuss more recent multimodal shortcut learning papers such as Dissecting Multimodality in VideoQA Transformer Models by Impairing Modality Fusion, ICML 2024.

**Limitations:**

Perhaps, the proposed method may be too complicated for some problems.

---

> ### Author Rebuttal · Authors · 2024-08-06
>
> Dear Reviewer 9n5E,
>
> We sincerely thank you for taking the time to review our paper. Our responses to your comments are provided here:
>
> ---
>
> **W1 “Theoretical justification that the encoder information can be disentangled to shared and unique representations”**:
>
> **A1:** To the best of our knowledge, common methods for disentangling shared and unique representations involve the use of estimators. The shared representation is learned by maximizing the cross-mutual information estimation while minimizing the mutual information between the shared and unique representations. In our paper, we minimize the SIC to maximize the lower bound of mutual information among shared representations. Similarly, our proposed self-penalization method utilizes self-supervised signals to tighten the lower bound, $I(Z_i, Z_j) \geq H^{\tilde{P}}(Z_j | Z_i) - H(Z_j | Z_i) + \log N - \widetilde{\mathcal{L}}_{\text{P-UIC}}$ (see Appendix D.3 for more details).
>
> Typically, mutual information is minimized by minimizing its upper bound, such as through adversarial objectives[1] or CLUB-like estimators[2,3]. However, these approaches are not equivalent to directly minimizing mutual information, despite many recent works striving to tighten the lower bound. These methods perform well in supervised tasks because task-relevant information can be well-defined (e.g., classification) and may not be suitable for pre training tasks. However, we focus more on how to preserve features in self-supervised learning. We conducted extensive experiments on the shortcuts dataset to confirm this: using only InfoNCE to fine-tune pre-trained models (CLIP, ResNet) on a new dataset results in the loss of a significant amount of original features, even though these features still exist in the shortcuts dataset (see Table 1).
>
> In the image-text self-supervised pre-training phase, we loosely define task-relevant information as $\mathcal{T} =\bigcup_{n=1}^N\{(x_{i1},...x_{ik})\cap(x_{j1},...x_{jm})\}_n$, where images and texts have a many-to-many relationship. Multiple texts can describe the same image (or video), with different texts holding both shared and unique information related to the image. Therefore, minimizing unique information between texts and images during the pre-training phase may not be appropriate. Hence, we choose UIC, which does not pull closer or push away the unique information of different modalities, but rather optimizes them within a plane to retain information that may be beneficial for downstream tasks (e.g., classification, segmentation).
>
> **Q1-2: “Shortcut learning in this case and discuss more recent multimodal shortcut learning papers”**
>
> **A2:** Shortcut learning, in the context of deep learning, refers to a phenomenon where neural networks exploit simple but suboptimal decision rules that work for most examples in the training set but fail to generalize to more challenging test examples [4]. This occurs when models learn to solve a task using features or heuristics that are correlated with the target in the training data but are not causally related to the task at hand [5]. In our case, under the setting of image-text retrieval, we synthesize images with MNIST image patches on them and append corresponding numbers to their associated captions. In this case, the model can easily get better performance by capturing MNIST shortcuts in the image and the corresponding numbers in the text under optimization of $\mathcal{L}_{\text{InfoNCE}}$, rather than focusing on more complex representational information present in both the image and text. And in downstream tasks when there is more complex unique representation in the data, the model fails and cannot complete the task well. More examples of synthesized shortcuts can be found in the submitted rebuttal PDF file.
>
> Besides, we are more than delighted to discuss with some recent work on multimodal shortcuts learning. Below are some brief discussion:  [6] uses the QUAG method to reveal that current VideoQA models often exploit dataset shortcuts rather than learning true multimodal representations. [7] proposes a novel backdoor attack method for multimodal-guided visual grasping systems leveraging shortcut learning and multimodal information. [8] provides comprehensive strategies for detecting and mitigating shortcut learning in VQA. Theses studies highlights the critical need to addressing shortcut learning in multimodal systems.
>
> **L1: “The proposed method may be too complicated for some problems”**
>
> **A3:** For some tasks, especially for tasks on some traditional datasets. With the rise of the pre-training era, we believe that for general models, capturing as much unique information as possible during the training phase may lead to better performance in downstream tasks. At the same time, we used an almost simplest version to experiment on more modalities (only introducing a small number of parameters, see Table 1 and Table 2), and this method can be plug-and-play to various architectures (ResNet, ViT, etc.). We believe that combining more advanced methods can further improve model performance.
>
> [1] Learning Disentangled Representations via Mutual Information Estimation. ECCV2020
>
> [2] CLUB: A Contrastive Log-ratio Upper Bound of Mutual Information. ICML2022.
>
> [3] Factorized contrastive learning: Going beyond multi-view redundancy. NeurIPS 2023.
>
> [4] Shortcut learning in deep neural networks." Nature Machine Intelligence 2020.
>
> [5] Can contrastive learning avoid shortcut solutions?. NeurIPS 2021.
>
> [6] Dissecting Multimodality in VideoQA Transformer Models by Impairing Modality Fusion. ICML2024.
>
> [7] Shortcut-enhanced Multimodal Backdoor Attack in Vision-guided Robot Grasping." Authorea Preprints 2024.
>
> [8] Shortcut Learning in Visual Question Answering.  2023.

---

> > ### Comment · Reviewer_9n5E · 2024-08-12
> >
> > Thanks for the responses. If possible, please include the discussion around theoretical justification in the paper and discuss related work. I have raised my rating to 7.

---

> > > ### Author Response · Authors · 2024-08-14
> > >
> > > Dear Reviewer 9n5E,
> > >
> > > We sincerely appreciate your recognition of our work. Thank you for your valuable feedback. We will incorporate more theoretical justification and a more comprehensive discussion of related work in subsequent versions of our paper.

---

### Official Review · Reviewer_KzFg · 2024-07-12

**Soundness:** 2
**Presentation:** 2
**Contribution:** 2
**Rating:** 7
**Confidence:** 3

**Summary:**

A new Multimodal Contrastive Learning method named QUEST is proposed to deal with the fine-grained alignment problem between different modal. Both quaternion contrastive objectives and orthogonal constraints are proposed to extract sufficient unique information. The quaternion vector spaces are designed to simultaneously optimize shared and unique information. Experiments conduct the superior performance in multimodal contrastive learning benchmarks.

**Strengths:**

The proposed method utilizes quadruple embedding to constrain unique information from different views in a plane space which avoid the degeneration of the unique decoder.
A self-penalization mechanism is proposed to penalize hard negative samples by dynamically re-weight the distribution of negative samples. And theoretical analysis is provided to show how this penalization effectively improves the extraction of unique information.
Experiments conduct the superior performance in multimodal contrastive learning benchmarks.

**Weaknesses:**

1) There are some typo, for example: line 218 MS-COCO-Cpation
2) The position of Z_i^s and Z_i^u of Modal M_i should be exchanged in Figure3. Also, Z_i_s and Z_i^u of Modal M_j should be Z_j^s and Z_j^u in Figure3.
3) Please check the position of + symbol in Eqn5.
4) Why the value of Eqn4 should be optimized to maximum value? Can you explain with an example in multimodal contrastive learning?

**Questions:**

Why the value of Eqn4 should be optimized to maximum value? Can you explain with an example in multimodal contrastive learning?

**Limitations:**

The experiments conducted in text and vision modal. Experiments on more modal data should be conducted to show the generalization ability.

---

> ### Author Rebuttal · Authors · 2024-08-06
>
> Dear Reviewer KzFg,
>
> We sincerely appreciate your thorough review and insightful comments. Please find our responses below.
>
> ------
>
> **W1-3: ”Presentations/Grammar/Typos”**
>
> **A1:** We apologize for the some typos in this paper and we have fixed it. We have re-examined Equation 5 and refined them. The notation $\mathbf z_j^{\mathbf n+}$ now correctly refers to the positive samples from modality $j$, and $\mathbf Z_{jk}^{\mathbf n-}$ refers to the negative samples from modality $j$. We appreciate your careful review and guidance.
>
> **W4: “Maximize the Equation 4 value”**
>
> **A2:** We apologize for any confusion caused by our oversight. In fact, we intend to maximize the absolute value of Equation 4 (Appendix.B3 for more details). We will correct this in the new version.
>
> First, we investigate why InfoNCE falls into shortcuts. We provide a simplified explanation in Equation 6: $-\frac{\partial\mathcal L_{\mathrm{InfoNCE}}}{\partial Z_{a}} = \frac{1}{\tau}(Z_{b}^{+} - \sum_{i=0}^{N}\beta_{i}Z_{bi})$. The term $\frac{1}{\tau}Z_b^+$ brings positive samples closer together, maximizing .  $||Z_{a}||\cdot||Z_{b}^+\sin \alpha$.However, in multi-view scenarios, $Z_{b}^{+}$ is sampled from $k$ positive samples as $Z_{b1}^{+}, Z_{b2}^{+}, \ldots, Z_{bk}^{+}$. Under the guidance of the gradient, the final representation tends to capture the shared information among all positive samples, leading to shortcuts.
>
> To capture unique information, we disentangle features into shared and unique representation. We apply strict constraints to bring the shared representations from different modalities closer together, while applying weaker constraints to keep the unique representations in the same plane (Intuitively, the unique information between different views is unrelated, we will not pull them closer) as shown in Figure 1(b). Maximizing the absolute value of Equation 4 is equivalent to optimizing the quaternion vectors $(\mathbf{Z}_i^\mathbf{s},\mathbf{Z}_i^\mathbf{u},\mathbf{Z}_j^\mathbf{s},\mathbf{Z}_j^\mathbf{u})$ to be in the same plane.
>
> ------
>
> **L1: ”Generalization ability for more modality”**
> **A3:**  We conducted additional experiments on common modalities including images, text, and audio.  Specifically, we performed image-audio experiments using the FMA and GTZAN datasets, as shown in Table 1. For text-audio experiments, we utilized the CLOTHO and AUDIOCAPS datasets, with results presented in Table 2. All of our models outperformed the baseline. Almost all of our models outperformed the baseline. For simplicity, we employed the almost simplest decoder structure (linear layers) and did not implement modality fusion or any cross-modal interactions. We believe that enhancing the architecture, strengthening cross-modal interactions, employing a larger batch size and incorporating pre-training would yield even higher performance.
>
> **Table 1: The results on the GTZAN and FMA datasets.**
>
> | Method  | Datasets | R@1       | R@5       | R@10      | R@1       | R@5       | R@10      |
> | ------- | -------- | --------- | --------- | --------- | --------- | --------- | --------- |
> |         | FMA      |   |       **i2a**     |           |   |       **a2i**     |           |
> | InfoNCE |          | 15.87     | 28.62     | 35.87     | 12.50     | 25.50     | 29.12     |
> | QUEST   |          | **17.83** | **30.87** | **38.50** | **13.50** | **26.52** | **30.62** |
> |         | GTZAN    |   |      **i2a**      |           | |      **a2i**        |           |
> | InfoNCE |          | 34.01     | 84.73     | 94.41     | 32.48     | 78.68     | 90.86     |
> | QUEST   |          | **41.62** | **88.83** | **97.65** | **35.53** | **82.23** | **93.40** |
>
> **Table 2: The results on the CLOTHO and AUDIOCAPS datasets.**
>
> | Method  | Datasets  | R@1       | R@5       | R@10      | R@1       | R@5       | R@10      |
> | ------- | --------- | --------- | --------- | --------- | --------- | --------- | --------- |
> |         | CLOTHO    |       |         **t2a**      |           |         |      **a2t**      |           |
> | InfoNCE |           | 20.16     | 51.30     | 66.56     | 20.06     | 52.66     | 68.23     |
> | QUEST   |           | **21.10** | **52.45** | **68.86** | **22.36** | **54.23** | **70.42** |
> |         | AUDIOCAPS |           |      **i2a**      |           |         |       **a2i**      |           |
> | InfoNCE |           | 4.59      | 17.79     | 26.22     | 5.45      | 15.78     | 22.48     |
> | QUEST   |           | **5.16**  | **18.08** | **27.17** | **6.02**  | **15.98** | **24.78** |

---

> > ### Comment · Reviewer_KzFg · 2024-08-13
> >
> > Thank you very much for the author's response. Most of my doubts have been resolved, and I have raised my score.

---

> > > ### Author Response · Authors · 2024-08-14
> > >
> > > Dear Reviewer KzFg,
> > >
> > > We extend our sincere gratitude for your thorough review of our work. We humbly accept the issues you've raised and  we will address them  in future iterations of our manuscript.
> > > Your insightful feedback is invaluable to us, and we are dedicated  to improving our work based on your suggestions.

---

### Author Rebuttal · Authors · 2024-08-06

Dear reviewers, we would like to sincerely thank all the reviewers for taking the time to read our paper and provide valuable feedback. We are delighted that reviewer KzFg (zJ9C) acknowledged the superior performance, innovation (9n5E, zJ9C), and reasonable motivation (9n5E, zJ9C) of our approach. Additionally, we appreciate all the reviewers' comments, which we have taken into consideration during the rebuttal period and made the following revisions:

1. According to reviewer KzFg's comments, we conducted experiments on additional modalities. To simplify, we did not employ any multimodal techniques (e.g., cross-modal interaction) and only added linear layers. Table 1 demonstrates that Quest(ours) achieved superior performance in the image-audio modality compared to the baseline model. Table 2 shows that Quest outperformed the baseline model in the text-audio modality. The experimental results indicate Quest's generalizability across more modality data.
2. In response to reviewers KzFg and 9n5E, we added examples of shortcuts in multimodal contrastive learning in Figure 1. Handwritten digits were simultaneously added to both text and image samples (with minimal impact on the original meaning of the images). However, training on these datasets led to a significant drop in model performance due to the tendency to learn easy features. Our model effectively addresses this issue. Even without adding these handwritten digits, an image is often described by multiple captions that usually hold different meanings. This is common in the real world and underscores the significance of our approach.
3. We appreciate reviewers KzFg and zJ9C for their suggestions regarding our phrasing and terminology. In our rebuttal, we provided explanations and committed to correcting all typos and terminology.

Please note that we have added tables and figures in the attached pdf to support our responses to the reviewers KzFg, 9n5E, and zJ9C.

---

### Decision · Program_Chairs · 2024-09-25

**Decision:**

Accept (poster)

**Comment:**

This paper introduces the Quadruple Multimodal Contrastive Learning Framework (QUEST), which is designed to extract both shared and unique task-relevant information across multiple modalities during contrastive learning. The framework is novel in its use of quaternion vector spaces to simultaneously optimize shared and unique information, thereby addressing the issue of "shortcut learning" that often arises in multimodal tasks. The method is empirically validated on several multimodal datasets, including popular public benchmarks such as Flickr30k and MSCOCO, where it demonstrates significant improvements in performance.

Given the significant strengths of the work, the thoughtful and comprehensive rebuttal, and the consensus among reviewers for acceptance, I recommend that this paper be accepted as a poster (as some reviewers yet noted a lack of rigorous theoretical analysis, particularly regarding the disentanglement of shared and unique information). The authors are encouraged to incorporate the suggestions for further theoretical exploration and to refine the presentation in the camera-ready version.